# DUAL GOAL REPRESENTATIONS

**Seohong Park**[*1]    **Deepinder Mann**[*1]    **Sergey Levine**[1]
[1]University of California, Berkeley
{seohong, dmann}@berkeley.edu

## ABSTRACT

In this work, we introduce **dual goal representations** for goal-conditioned reinforcement learning (GCRL). A dual goal representation characterizes a state by "the set of temporal distances from all other states"; in other words, it encodes a state through its *relations* to every other state, measured by temporal distance. This representation provides several appealing theoretical properties. First, it depends only on the intrinsic dynamics of the environment and is invariant to the original state representation. Second, it contains provably sufficient information to recover an optimal goal-reaching policy, while being able to filter out exogenous noise. Based on this concept, we develop a practical goal representation learning method that can be combined with any existing GCRL algorithm. Through diverse experiments on the OGBench task suite, we empirically show that dual goal representations consistently improve offline goal-reaching performance across 20 state- and pixel-based tasks.

Blog post: https://seohong.me/blog/dual-representations/

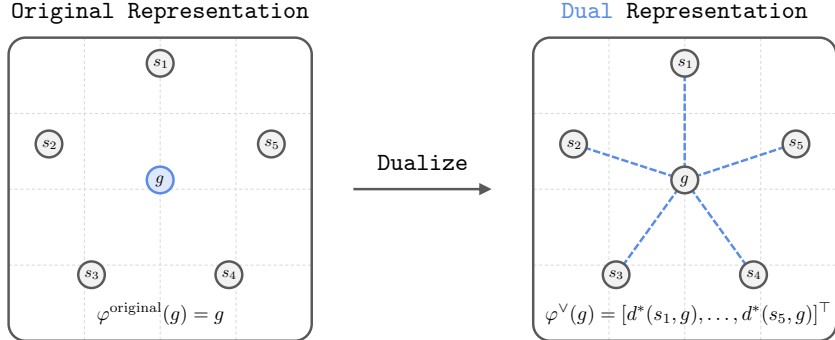

Figure 1: **Dual goal representations.** A dual goal representation $\varphi^\vee(g)$ is defined as the set of temporal distances $d^*(\cdot, g)$ from all other states. This representation has a number of appealing properties: it only depends on the intrinsic dynamics of the environment, contains sufficient information to express an optimal goal-reaching policy, and is able to discard exogenous noise. In continuous environments, we approximate the dual representation $\varphi^\vee(g)$ by the goal embedding $\varphi(g)$ in a parameterized temporal distance function, $d^*(s, g) = \psi(s)^\top \varphi(g)$.

## 1 INTRODUCTION

Representation learning lies at the heart of modern machine learning. The successes of large-scale unsupervised pre-training (Brown et al., 2020; Ramesh et al., 2022) have shown that rich representations can enable both efficient adaptation and strong generalization across tasks. In this work, we study representation learning for an important class of reinforcement learning (RL) problems, goal-conditioned reinforcement learning (GCRL) (Kaelbling, 1993; Park et al., 2025a). The objective of GCRL is to train a multi-task policy capable of reaching any target state from any starting state in minimal time. Many real-world tasks, including navigation, object manipulation, and even games, naturally fit into this goal-reaching framework.

An important question in GCRL is how to represent goals. While many works simply use the original state observations provided by the environment to represent goal states (Kaelbling, 1993;

---
*Equal contribution.

[Park et al., 2025a](#)), these observations are not necessarily best structured to describe goals (*i.e.*, tasks), and often contain superfluous information, such as uncontrollable noise. On the other hand, a good, well-designed goal representation can accelerate policy learning during training, improve goal-reaching performance at test time, and even potentially enable robust generalization to out-of-distribution goals.

To achieve these benefits, we define two desiderata for an ideal goal representation. First, it must be able to capture sufficient information to represent an optimal goal-conditioned policy (*sufficiency*). Second, it must be able to discard irrelevant information, such as background noise, that does not affect goal-reaching (*noise invariance*).

Our key idea to achieve these desiderata is to represent a goal as **the set of temporal distances from all other states**. More specifically, we represent a goal as a *functional* (*i.e.*, a function from the state space to $\mathbb{R}$) that takes a state as input and outputs the corresponding temporal distance. We call this a **dual goal representation**. We theoretically show that this dual goal representation satisfies both desiderata (sufficiency and noise invariance): *i.e.*, we can represent an optimal policy solely based on the dual representation, and the representation is invariant to exogenous noise.

Building on this theoretical insight, we propose a new practical goal representation learning recipe for offline goal-conditioned RL. Our method first trains a parameterized goal-conditioned value function with an offline value learning algorithm to compute temporal distances. Then, we take the goal embedding of the learned value function as an approximate representation of the functional, which is then used as a goal representation for the policy. We demonstrate that this goal representation learning recipe consistently improves the performance and generalizability of goal-conditioned policies, across diverse robotic locomotion and manipulation environments.

**Contributions.** Our main contributions are the theoretical formulation and empirical verification of dual goal representations. First, we theoretically show that dual goal representations can learn a sufficient and noise-invariant representation for optimal goal reaching. Second, we demonstrate that our empirical recipe based on this principle improves performance and generalization capabilities in practice, outperforming previous goal representation learning approaches on 20 diverse goal-reaching tasks in robotic locomotion and manipulation.

## 2 PRELIMINARIES

**Controlled Markov processes and goal-conditioned RL.** Let $\mathcal{S}$ be a state space, $\mathcal{A}$ be an action space, and $p(s' \mid s, a) : \mathcal{S} \times \mathcal{A} \to \Delta(\mathcal{S})$ be a transition dynamics kernel, where $\Delta(\mathcal{X})$ denotes the set of valid probability distributions over a space $\mathcal{X}$ and placeholder variables are denoted in gray. For theoretical results, we assume that $\mathcal{S}$ and $\mathcal{A}$ are discrete spaces. This is only for the sake of simplicity in discussion, and our results can be extended to continuous state spaces with appropriate modifications.

A controlled Markov process (CMP) is defined by the tuple $\mathcal{M} = (\mathcal{S}, \mathcal{A}, p)$. Given a goal-conditioned policy $\pi(a \mid s, g) : \mathcal{S} \times \mathcal{S} \to \Delta(\mathcal{A})$, and a goal-conditioned reward function $r(s, g) : \mathcal{S} \times \mathcal{S} \to \mathbb{R}$, we define $V^\pi(s, g) = \mathbb{E}_{\tau \sim p^\pi(\tau \mid s_0 = s, g)}[\sum_{t=0}^\infty \gamma^t r(s_t, g)]$ and $Q^\pi(s, a, g) = \mathbb{E}_{\tau \sim p^\pi(\tau \mid s_0 = s, a_0 = a, g)}[\sum_{t=0}^\infty \gamma^t r(s_t, g)]$, where $\gamma \in (0, 1)$ denotes a discount factor and $p^\pi$ denotes the trajectory distribution induced by the policy and transition dynamics. Unless otherwise mentioned, we define $r(s, g) = \mathbb{I}(s = g)$, where $\mathbb{I}$ is the 0-1 indicator function, and assume that the agent enters an absorbing state once it reaches the goal (*i.e.*, it can get a reward of 1 at most once). We define the optimal value functions as $V^*(s, g) = \max_{\pi : \mathcal{S} \times \mathcal{S} \to \Delta(\mathcal{A})} V^\pi(s, g)$ and $Q^*(s, a, g) = \max_{\pi : \mathcal{S} \times \mathcal{S} \to \Delta(\mathcal{A})} Q^\pi(s, a, g)$. We also define the *temporal distance* function as $d^*(s, g) = \log_\gamma V^*(s, g)$. Note that, in deterministic environments, $d^*(s, g)$ corresponds to the length of a shortest path from $s$ to $g$.

The objective of offline goal-conditioned RL is to find a goal-conditioned policy $\pi$ that maximizes $V^\pi(s, g)$ for all $s, g \in \mathcal{S}$, from a dataset $\mathcal{D} = \{\tau^{(i)}\}_{i=1}^M$ consisting of state-action trajectories $\tau = (s_0, a_0, s_1, \ldots, s_T)$ with no environment interactions.

**Block CMPs.** Some theoretical results in this work are built upon the block CMP (BCMP) framework ([Du et al., 2019](#); [Efroni et al., 2022](#); [Lamb et al., 2022](#)). A BCMP is defined by a tuple $(\mathcal{S}, \mathcal{Z}, \mathcal{A}, p, p^e, p^d)$, where $\mathcal{S}$ is an observation space, $\mathcal{Z}$ is a latent state space, $\mathcal{A}$ is an action space,

$p(z' \mid z, a) : \mathcal{Z} \times \mathcal{A} \to \mathcal{Z}$ is a latent transition dynamics kernel, $p^e(s \mid z) : \mathcal{Z} \to \Delta(\mathcal{S})$ is an observation emission distribution, and $p^\ell(s) : \mathcal{S} \to \mathcal{Z}$ is a latent mapping function, such that $p^\ell(s) = z$ for all $z \in \mathcal{Z}$ and $s \sim p^e(s \mid z)$. Intuitively, a BCMP can be thought of as a latent CMP $(\mathcal{Z}, \mathcal{A}, p)$ augmented with noisy observations, such that the observation distributions from different latent states always have *disjoint* supports (*i.e.*, every observation maps to a unique latent state).

In a BCMP, we define a goal-conditioned reward function as $r^\ell(s, g) = \mathbb{I}(p^\ell(s) = p^\ell(g))$. Note that this only depends on the latent states. Given a goal-conditioned policy, $\pi(a \mid s, g) : \mathcal{S} \times \mathcal{S} \to \Delta(\mathcal{A})$, we define the value and Q functions and temporal distance function similarly as before, but with the modified reward function $r^\ell$. The BCMP framework encompasses many real-world scenarios (Efroni et al., 2022; Lamb et al., 2022), including image-based robotic control (where latent states correspond to the ground-truth poses of the robot arm and objects, and the emission function corresponds to a noisy yet fully observable rendering of the current state).

## 3 DUAL GOAL REPRESENTATIONS

The main aim of this work is to find an effective goal representation $\varphi(g) : \mathcal{S} \to \mathcal{W}$ for a goal-conditioned policy $\pi(a \mid s, \varphi(g))$ parameterized by $\varphi$, where $\mathcal{W}$ denotes a latent representation space. A good goal representation can offer several benefits. During training, it can reduce the learning complexity of the policy network $\pi$ by providing structured information about the goal state. During evaluation, it can help the agent generalize to unseen, potentially out-of-distribution goals.

What constitutes an ideal goal representation? We define two desiderata. First, it must be able to capture enough information about the state to represent an optimal policy (*sufficiency*). Second, it must be able to discard information irrelevant to goal-reaching (*noise invariance*). In other words, an ideal goal representation should be able to maintain only a necessary *and* sufficient amount of information to perform optimal goal-reaching. This will make the agent focus on only the controllable part of the state, enabling faster policy learning and better generalization.

### 3.1 THE IDEA

Our main idea to achieve these desiderata is to represent a goal as **the set of the temporal distances from all other states**. For example, in a discrete state space $\mathcal{S} = \{s_1, s_2, \ldots, s_K\}$, this corresponds to the vector consisting of all temporal distances to that goal; *i.e.*, $\varphi(g) = [d^*(s_1, g), d^*(s_2, g), \ldots, d^*(s_K, g)]^\top$. In the general case, this representation corresponds to a *function* from $\mathcal{S}$ to $\mathbb{R}$ (such a function is called a *functional*).

Formally, for a CMP $\mathcal{M} = (\mathcal{S}, \mathcal{A}, p)$, we define $\varphi^\vee : \mathcal{S} \to (\mathcal{S} \to \mathbb{R})$ as

$$\varphi^\vee(g) = (s \mapsto d^*(s, g))$$

for $s, g \in \mathcal{S}$. In other words, $\varphi^\vee(g)(s) = d^*(s, g)$. We call $\varphi^\vee$ the **dual goal representation** (or simply dual representation) function of the CMP $\mathcal{M}$.

Intuitively, dual goal representations replace a goal state with its *relations* to all other states, measured by temporal distance. This has several benefits. First, the resulting representation is *invariant* to the original state representation given by the environment, as it depends only on the intrinsic temporal dynamics of the environment. Second, it can be shown that this representation retains sufficient information to represent an optimal goal-reaching policy, while being able to discard exogenous noise, as we formalize in the following section.

### 3.2 THEORETICAL PROPERTIES

We now state two important theoretical properties of dual goal representations. The first property is sufficiency:

**Theorem 3.1** (Sufficiency of Dual Goal Representations)**.** *Let $\mathcal{M} = (\mathcal{S}, \mathcal{A}, p)$ be a CMP and $\varphi^\vee$ be its dual goal representation function. Then, there exists a deterministic policy $\pi^\vee : \mathcal{S} \times \mathcal{S}^\vee \to \mathcal{A}$ that takes a dual goal representation as input, such that its induced policy $\tilde{\pi} : \mathcal{S} \times \mathcal{S} \to \mathcal{A}$ defined as $\tilde{\pi}(s, g) := \pi^\vee(s, \varphi^\vee(g))$ satisfies $V^{\tilde{\pi}}(s, g) = V^*(s, g)$ for all $s, g \in \mathcal{S}$.*

The proof can be found in Section A. Intuitively, the above theorem states that the dual goal representation function $\varphi^\vee$ retains enough information so that one can recover an optimal policy that depends only on dual goal representations. The second property is noise invariance:

> **Digressive remark:** Connections to other concepts in mathematics and machine learning
>
> The dual goal representation is based on the philosophy of a "relative point of view" (*i.e.*, understanding an object via its relation to all other objects). This is analogous to various concepts in mathematics and machine learning, and we highlight a few examples. In category theory, the *Yoneda lemma* implies that we can identify an object through its morphisms to all other objects in the same category (Riehl, 2017). In kernel machines and functional analysis, the *Riesz representation theorem* implies that we can uniquely represent a point in a Hilbert space as a linear functional in its dual space (Folland, 1999; Wainwright, 2019). The dual goal representation is named after this analogy.

**Theorem 3.2** (Noise Invariance of Dual Goal Representations). *Let $\mathcal{M} = (\mathcal{S}, \mathcal{Z}, \mathcal{A}, p, p^e, p^\ell)$ be a BCMP and $\varphi^\vee$ be its dual goal representation function. Let $g_1, g_2 \in \mathcal{S}$ be two goal observations from the same latent state; i.e., $p^\ell(g_1) = p^\ell(g_2)$. Then, they have the same dual goal representation; i.e., $\varphi^\vee(g_1) = \varphi^\vee(g_2)$.*

Again, the proof can be found in Section A. Intuitively, this theorem states that, under some technical assumptions (based on the BCMP framework described in Section 2), dual goal representations are invariant under exogenous noise in the observation space. Combining Theorems 3.1 and 3.2, we conclude that dual goal representations satisfy both desiderata described at the beginning of Section 3.

## 4 PRACTICAL INSTANTIATION

There are two challenges with implementing a practical learning algorithm for dual goal representations. First, the functional form of dual goal representations cannot be directly implemented in practice, unless the state space is small and discrete (in which case we can represent a function as a finite-dimensional vector). Second, we need to know the temporal distance function $d^*$.

**Approximating functionals.** We address the first challenge by employing a *parameterized* temporal distance function. Namely, we model the temporal distance function as follows:

$$d^*(s, g) = f(\psi(s), \varphi(g)), \tag{1}$$

where $\psi(s) : \mathcal{S} \to \mathbb{R}^N$ and $\varphi(g) : \mathcal{S} \to \mathbb{R}^N$ are respectively the state and goal embeddings, and $f : \mathbb{R}^N \times \mathbb{R}^N \to \mathbb{R}$ is a (simple) aggregation function that combines the information from both embeddings to compute the temporal distance. Then, we treat the output of the goal embedding, $\varphi(g)$, as a practical finite-dimensional representation of the functional dual goal representation, $\varphi^\vee(g)$. Intuitively, this makes $\varphi$ capture temporal information about the goal, in the sense that we can compute the temporal distance by pairing with any other state $s$ via $f(\psi(s), \varphi(g))$.

For the aggregation function $f$, we use the following inner product parameterization:

$$f(\psi(s), \varphi(g)) = \psi(s)^\top \varphi(g). \tag{2}$$

We note that this inner product form is *universal* (Park et al., 2024b), meaning that it is expressive enough to approximate any two-variable function (on a compact set) up to an arbitrary accuracy.

**Approximating temporal distance.** Next, we approximate the temporal distance function $d^*$ in practice, using an existing offline goal-conditioned value learning algorithm. In particular, we employ goal-conditioned IQL (Kostrikov et al., 2022; Park et al., 2025a) to approximate $d^*$ (to be precise, a transformed variant of $d^*$ via $V^*$, as we explain below). While dual goal representations can be instantiated with any other goal-conditioned value learning (or temporal distance learning) algorithm, we find that goal-conditioned IQL generally leads to the best performance while remaining simple.

Specifically, goal-conditioned IQL trains a parameterized goal-conditioned value function $V(s, g) = f(\psi(s), \varphi(g)) : \mathcal{S} \times \mathcal{S} \to \mathbb{R}$ and a goal-conditioned Q function $Q(s, a, g) : \mathcal{S} \times \mathcal{A} \times \mathcal{S} \to \mathbb{R}$ with the following losses (Kostrikov et al., 2022):

$$L(\psi, \varphi) = \mathbb{E}_{s, a, g \sim \mathcal{D}} \left[ \ell_\kappa^2 (f(\psi(s), \varphi(g)) - \bar{Q}(s, a, g)) \right], \tag{3}$$

$$L(Q) = \mathbb{E}_{s, a, s', g \sim \mathcal{D}} \left[ (Q(s, a, g) - r(s, g) - \gamma f(\psi(s'), \varphi(g)))^2 \right], \tag{4}$$

---

**Algorithm 1** Offline Goal-Conditioned RL with Dual Goal Representations

---

▷ Dual goal representation learning
Initialize state representation $\psi(s)$, goal representation $\varphi(g)$, Q function $Q(s, a, g)$
**while** not converged **do**

  Sample batch $\{(s, a, s', g)^{(i)}\}_i$ from $\mathcal{D}$

  ▷ Train parameterized value function $f(\psi(s), \varphi(g))$ with goal-conditioned IQL
  Train $\psi$, $\varphi$ by minimizing $\mathbb{E}[\ell_\kappa^2(f(\psi(s), \varphi(g)) - \bar{Q}(s, a, g))]$
  Train $Q$ by minimizing $\mathbb{E}[(Q(s, a, g) - r(s, g) - \gamma f(\psi(s'), \varphi(g)))^2]$
  Update $\bar{Q}$ using exponential moving averaging

▷ Downstream offline GCRL with dual goal representation (can be run in parallel with above)
Initialize policy $\pi(a \mid s, \varphi(g))$
(If necessary) initialize representation-conditioned value functions $V^{\mathrm{GCRL}}(s, \varphi(g))$, $Q^{\mathrm{GCRL}}(s, a, \varphi(g))$
**while** not converged **do**

  Train $\pi$, $V^{\mathrm{GCRL}}$, $Q^{\mathrm{GCRL}}$ with any offline GCRL algorithm (*e.g.*, GCBC, GCIVL, CRL)
**return** $\pi(a \mid s, \varphi(g))$

---

where $\ell_\kappa^2$ denotes the expectile loss with expectile $\kappa$ (Kostrikov et al., 2022) and $\bar{Q}$ denotes the target Q network (Mnih et al., 2013).

If the reward function is given as the 0-1 indicator function, $r(s, g) = \mathbb{I}(s = g)$, we have the relation $V^*(s, g) = \gamma^{d^*(s,g)}$ (Section 2). Hence, when the IQL losses converge, $f(\psi(s), \varphi(g)) = \psi(s)^\top \varphi(g)$ approximates this transformed temporal distance function, $V^*(s, g) = \gamma^{d^*(s,g)}$. In practice, we use the $-1$-0 reward function (*i.e.*, $r(s, g) = \mathbb{I}(s = g) - 1$) instead of the 0-1 indicator function, following Park et al. (2025a). While this leads to a slightly different relation $V^*(s, g) = -(1 - \gamma^{d^*(s,g)})/(1 - \gamma)$, we found this version to work better in practice. Note that regardless of the choice of reward function, $f(\psi(s), \varphi(g))$ still approximates a (transformed) temporal distance function.

**Downstream offline goal-conditioned RL.** After obtaining a dual goal representation $\varphi$, we train a parameterized goal-conditioned policy $\pi(a \mid s, \varphi(g))$ (and optionally goal-conditioned value functions, $V^{\mathrm{GCRL}}(s, \varphi(g))$ and $Q^{\mathrm{GCRL}}(s, a, \varphi(g))$) with an existing offline GCRL algorithm to perform downstream goal-conditioned policy learning. Note that the choice of this downstream GCRL algorithm is orthogonal to the one used to approximate the temporal distance in the previous paragraph, and thus our method can be applied to any general GCRL algorithm. In this work, we apply dual goal representations to three different offline GCRL algorithms, goal-conditioned IVL (GCIVL) (Kostrikov et al., 2022; Park et al., 2025a), contrastive RL (CRL) (Eysenbach et al., 2022), and goal-conditioned flow BC (GCFBC) (Park et al., 2025b). We describe the full goal-conditioned RL procedure in Algorithm 1, where $\ell_\kappa^2$ denotes the expectile loss with expectile $\kappa$ (Kostrikov et al., 2022) and $\bar{Q}$ denotes the target Q network (Mnih et al., 2013).

## 5 RELATED WORK

**Offline goal-conditioned RL.** At the intersection of offline RL (Lange et al., 2012; Levine et al., 2020) and goal-conditioned RL (Kaelbling, 1993), offline goal-conditioned RL aims to train a goal-reaching policy from an unlabeled (*i.e.*, reward-free) dataset. Offline GCRL provides a principled way to pre-train policies, representations, and value functions in a self-supervised manner, which can later be adapted to downstream tasks (Ma et al., 2023; Ghosh et al., 2023; Kim et al., 2024; Park et al., 2025a). Prior works in offline goal-conditioned RL have proposed diverse approaches based on implicit value learning (Park et al., 2023), contrastive learning (Eysenbach et al., 2022; Zheng et al., 2024), metric learning (Wang et al., 2023; Myers et al., 2024), and planning (Savinov et al., 2018; Eysenbach et al., 2019; Wang et al., 2024). In this work, we aim to improve the performance and generalizability of goal-reaching agents by learning an effective goal representation (as with prior goal representation learning methods discussed below), which can be orthogonally combined with any existing offline GCRL algorithm.

**Representation learning for RL.** Representation learning has long been studied in RL (Echchahed & Castro, 2025). By learning structured representations for states, actions, or other components in the given environment, prior work aims to facilitate learning and improve the generalization of RL

agents. To this end, a variety of representation learning techniques (Li et al., 2006; Castro, 2019; Wu et al., 2019; Laskin et al., 2020; Schwarzer et al., 2021) have been proposed, and we refer to Echchahed & Castro (2025) for a comprehensive literature review. Among representation learning algorithms, our practical method has a resemblance to the successor representations and successor features (Dayan, 1993; Barreto et al., 2017; Touati & Ollivier, 2021; Touati et al., 2023). Unlike these works, we approximate *optimal* temporal distances (or equivalently *optimal* goal-conditioned values), which do not correspond to any successor features or representations with respect to a fixed policy. This enables us to directly learn a representation for optimal goal-reaching.

**Representation learning for GCRL.** Our work is most closely related to previous studies that propose representation learning techniques based on goal-conditioned RL (Sermanet et al., 2018; Eysenbach et al., 2022; Steccanella & Jonsson, 2022; Ma et al., 2023; Park et al., 2024a; Myers et al., 2025; Lawson et al., 2025). While the concept of dual goal representations and our theoretical results (Section 3) are, to our knowledge, novel, our practical learning algorithm for dual goal representations (Section 4) bears similarity to several existing algorithms. Some previous works train temporal distance representations $\varphi$ using the metric parameterization $\|\varphi(s) - \varphi(g)\|_2$ with different metric learning algorithms (Sermanet et al., 2018; Steccanella & Jonsson, 2022; Ma et al., 2023; Park et al., 2024a), which can potentially be viewed as variants of dual goal representations with a different aggregation function. Our method differs from these works in two ways. First, we use $\varphi$ for *goal representations* (as our theory in Section 3 suggests), whereas these works use $\varphi$ for metric-based skill learning (Park et al., 2024a), state representations (Ma et al., 2023), reward shaping (Sermanet et al., 2018; Steccanella & Jonsson, 2022; Ma et al., 2023), and planning (Sermanet et al., 2018; Steccanella & Jonsson, 2022; Ma et al., 2023; Park et al., 2024a). Second, unlike these works, we employ the more general inner product parameterization (instead of the metric one), which we show leads to better performance likely due to its universality. Similar to our work, TRA (Myers et al., 2025) employs the inner product parameterization to learn a goal representation. However, this method trains the representation with *behavioral* temporal contrastive learning, which does not aim to approximate $d^*$; in contrast, we train a representation based on the *optimal* value function (as our theory suggests), and we empirically show that our method leads to substantially better performance in Section 6.

## 6 EXPERIMENTS

In this section, we empirically verify the effectiveness of dual goal representations through a series of experiments. In Section 6.1, we evaluate the performance of "ideal" dual representations (Section 3) on discrete-space tasks as a proof of concept. Next, we evaluate our practical goal representation learning algorithm (Section 4) on a standard offline goal-conditioned RL benchmark (Park et al., 2025a), comparing with various previous representation learning approaches. Then, we provide ablation studies of our method in Section 6.3 to understand the importance of each component.

In our experiments, we use 8 random seeds for the main result table (Table 1) and 4 seeds for other tables and plots, unless otherwise stated. We report standard deviations in tables and 95% confidence intervals in plots. In tables, we highlight numbers that are at and above 95% of the best performance, following Park et al. (2025a).

### 6.1 RESULTS WITH "IDEAL" DUAL REPRESENTATIONS

We begin our experiments by demonstrating how our original concept of dual representations (*i.e.*, the "ideal" dual representation described in Section 3, not the approximated one in Section 4) can potentially improve goal-reaching capabilities. To this end, we consider a discrete puzzle environment where we can analytically precompute the temporal distances $d^*$. Specifically, we employ the discrete "Lights Out" puzzle (Park et al., 2025a).

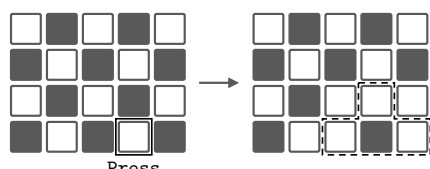

Figure 2: **The "Lights Out" puzzle.**

The aim of this puzzle is to reach a given goal configuration from an initial state, where pressing a button toggles the colors of that button and the adjacent ones (Figure 2). The dataset is collected by a uniform random policy, and we evaluate agents with five pre-defined state-goal pairs on two different puzzle sizes, 4x5 and 4x6 (Park et al., 2025a).[1]

---

[1]Note that these puzzle tasks are tabular variants of the ones in OGBench (Park et al., 2025a).

On these tasks, we compare the performance of the original and dual representations. They are both trained with goal-conditioned DQN (Mnih et al., 2013). For dual representations, to reduce dimensionality, we randomly sample 64 states $\{s_i\}_{i=1}^{64}$ (*i.e.*, 64 random board configurations), and use $[d^*(s_1, g), \ldots, d^*(s_{64}, g)]^\top$ as the representation of $g$. In this experiment, we use the dual representation for both states and goals. We present the comparison result in Figure 3. The result suggests that our dual representations substantially improve training speed and goal-reaching performance. Note that even though the dual representation $\varphi^\vee$ is *bijective* in this case, it provides structured (processed) information about the temporal dynamics, which makes policy learning significantly easier. This highlights the benefits of using a structured representation.

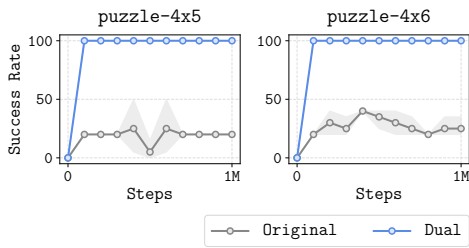

Figure 3: **Original vs. dual representations.**

**Disclaimer:** In this experiment, our goal is to show how dual representations improve performance in the *ideal* case (as a proof of concept), and we use privileged information about ground-truth temporal distances. In the following section, we demonstrate the performance of practical dual goal representations without such privileged information.

## 6.2 RESULTS ON OGBENCH

Next, we evaluate our practical algorithm for dual goal representations (Algorithm 1) on OG-Bench (Park et al., 2025a), a benchmark suite for offline goal-conditioned RL algorithms.

**Tasks and datasets.** We employ 13 state-based and 7 pixel-based tasks from OGBench across robotic navigation and manipulation (Figure 5). In navigation tasks ({point, ant, humanoid}maze), the aim is to control a robot body (from a single point to a complex humanoid robot with 21 degrees of freedom) to reach a desired goal location in a maze. In antsoccer, the agent must control a quadrupedal robot to dribble a soccer ball to a desired location. In manipulation tasks (cube, scene, puzzle), the agent must control a robotic arm to manipulate cubes, re-arrange everyday objects, or solve a combinatorial puzzle. In pixel-based environments, the agent must perform robotic manipulation solely from $64 \times 64 \times 3$-sized image observations. We use the standard navigate or play datasets for these tasks, where the datasets are collected by task-agnostic "play"-style (Lynch et al., 2019) demonstrations that randomly perform diverse atomic behaviors (*e.g.*, continuously reaching random locations or performing random button presses).

**Previous goal representation learning algorithms.** We compare dual goal representations with five previous goal representation learning methods: Original, VIB (Shah et al., 2021; Park et al., 2023), VIP (Ma et al., 2023), TRA (Myers et al., 2025), and BYOL-$\gamma$ (Lawson et al., 2025). Original ("Orig") is a baseline that does not use any representations (*i.e.*, $\varphi(g) = g$). VIB (Shah et al., 2021; Park et al., 2023) trains goal representations via a variational information bottleneck (VIB) within a goal-conditioned value function or policy. VIP (Ma et al., 2023) trains metric-based goal representations with a goal-conditioned value learning algorithm. TRA (Myers et al., 2025) trains contrastive goal representations with behavioral contrastive learning (Eysenbach et al., 2022). BYOL-$\gamma$ (Lawson et al., 2025) trains self-predictive goal representations with temporal self-supervised learning. We refer to Section D for the full implementation details about these methods.

**Downstream offline GCRL algorithms.** We measure the effectiveness of goal representations by training a parameterized goal-conditioned policy $\pi(a \mid s, \varphi(g))$ with downstream offline GCRL algorithms, as described in Algorithm 1. To extensively assess the performance of goal representations, we consider three downstream algorithms across diverse categories: goal-conditioned implicit V-learning (GCIVL) (Kostrikov et al., 2022; Park et al., 2025a), contrastive RL (CRL) (Eysenbach et al., 2022), and goal-conditioned flow behavioral cloning (GCFBC) (Park et al., 2025b). We refer to Section D for full details.

### 6.2.1 RESULTS ON STATE-BASED TASKS

We report the performance of six goal representation learning methods combined with three downstream GCRL algorithms (18 combinations in total) on 13 state-based OGBench environments in Table 1 above and Tables 6 and 7 in Section C. These results suggest that our dual goal representations achieve the best average performance in all three settings, outperforming both the previous

Table 1: **Results on state-based tasks with GCIVL as the downstream algorithm.** Our dual goal representation mostly achieves the best performance among goal representation learning methods.

| Environment | Orig | VIB | VIP | TRA | BYOL-$\gamma$ | Dual |
|---|---|---|---|---|---|---|
| pointmaze-medium-navigate-v0 | 78 $_{\pm 8}$ | 69 $_{\pm 13}$ | 0 $_{\pm 1}$ | 3 $_{\pm 6}$ | 37 $_{\pm 7}$ | 76 $_{\pm 7}$ |
| pointmaze-large-navigate-v0 | 52 $_{\pm 6}$ | 50 $_{\pm 7}$ | 0 $_{\pm 0}$ | 1 $_{\pm 2}$ | 22 $_{\pm 12}$ | 46 $_{\pm 6}$ |
| antmaze-medium-navigate-v0 | 71 $_{\pm 4}$ | 68 $_{\pm 4}$ | 31 $_{\pm 5}$ | 22 $_{\pm 15}$ | 39 $_{\pm 5}$ | 75 $_{\pm 4}$ |
| antmaze-large-navigate-v0 | 16 $_{\pm 3}$ | 9 $_{\pm 3}$ | 9 $_{\pm 2}$ | 22 $_{\pm 12}$ | 11 $_{\pm 5}$ | 28 $_{\pm 11}$ |
| antmaze-giant-navigate-v0 | 0 $_{\pm 0}$ | 0 $_{\pm 0}$ | 0 $_{\pm 0}$ | 0 $_{\pm 0}$ | 0 $_{\pm 0}$ | 0 $_{\pm 0}$ |
| humanoidmaze-medium-navigate-v0 | 27 $_{\pm 3}$ | 24 $_{\pm 2}$ | 7 $_{\pm 3}$ | 21 $_{\pm 3}$ | 18 $_{\pm 5}$ | 29 $_{\pm 3}$ |
| humanoidmaze-large-navigate-v0 | 3 $_{\pm 0}$ | 3 $_{\pm 1}$ | 1 $_{\pm 0}$ | 2 $_{\pm 1}$ | 2 $_{\pm 1}$ | 3 $_{\pm 2}$ |
| antsoccer-arena-navigate-v0 | 47 $_{\pm 4}$ | 34 $_{\pm 4}$ | 2 $_{\pm 1}$ | 8 $_{\pm 2}$ | 11 $_{\pm 4}$ | 31 $_{\pm 3}$ |
| cube-single-play-v0 | 52 $_{\pm 3}$ | 90 $_{\pm 3}$ | 40 $_{\pm 7}$ | 40 $_{\pm 5}$ | 51 $_{\pm 11}$ | 89 $_{\pm 3}$ |
| cube-double-play-v0 | 35 $_{\pm 5}$ | 33 $_{\pm 3}$ | 3 $_{\pm 2}$ | 7 $_{\pm 2}$ | 6 $_{\pm 4}$ | 60 $_{\pm 4}$ |
| scene-play-v0 | 46 $_{\pm 3}$ | 58 $_{\pm 1}$ | 23 $_{\pm 6}$ | 46 $_{\pm 6}$ | 44 $_{\pm 9}$ | 72 $_{\pm 6}$ |
| puzzle-3x3-play-v0 | 5 $_{\pm 1}$ | 14 $_{\pm 3}$ | 3 $_{\pm 1}$ | 5 $_{\pm 1}$ | 0 $_{\pm 0}$ | 5 $_{\pm 1}$ |
| puzzle-4x4-play-v0 | 14 $_{\pm 1}$ | 6 $_{\pm 3}$ | 1 $_{\pm 1}$ | 10 $_{\pm 3}$ | 1 $_{\pm 2}$ | 23 $_{\pm 3}$ |
| Average | 34 $_{\pm 1}$ | 35 $_{\pm 2}$ | 9 $_{\pm 1}$ | 15 $_{\pm 2}$ | 19 $_{\pm 2}$ | 41 $_{\pm 2}$ |

Table 2: **Results on pixel-based tasks.** Dual goal representations achieve the best performance on five out of seven pixel-based tasks. All goal representation learning methods struggle with the `puzzle` tasks, likely due to the use of "late fusion" (see Section 6.2.2).

| Environment | Orig | VIB | VIP | TRA | BYOL-$\gamma$ | Dual |
|---|---|---|---|---|---|---|
| visual-antmaze-medium-navigate-v0 | 66 $_{\pm 4}$ | 18 $_{\pm 9}$ | 30 $_{\pm 7}$ | 48 $_{\pm 4}$ | 32 $_{\pm 5}$ | 78 $_{\pm 4}$ |
| visual-antmaze-large-navigate-v0 | 26 $_{\pm 5}$ | 5 $_{\pm 2}$ | 9 $_{\pm 1}$ | 13 $_{\pm 3}$ | 9 $_{\pm 4}$ | 40 $_{\pm 4}$ |
| visual-cube-single-play-v0 | 53 $_{\pm 4}$ | 18 $_{\pm 19}$ | 39 $_{\pm 6}$ | 31 $_{\pm 24}$ | 35 $_{\pm 8}$ | 58 $_{\pm 5}$ |
| visual-cube-double-play-v0 | 9 $_{\pm 2}$ | 0 $_{\pm 0}$ | 0 $_{\pm 0}$ | 3 $_{\pm 2}$ | 2 $_{\pm 1}$ | 9 $_{\pm 2}$ |
| visual-scene-play-v0 | 25 $_{\pm 2}$ | 6 $_{\pm 3}$ | 4 $_{\pm 1}$ | 15 $_{\pm 6}$ | 10 $_{\pm 8}$ | 26 $_{\pm 5}$ |
| visual-puzzle-3x3-play-v0 | 22 $_{\pm 2}$ | 0 $_{\pm 0}$ | 0 $_{\pm 0}$ | 0 $_{\pm 0}$ | 0 $_{\pm 0}$ | 0 $_{\pm 0}$ |
| visual-puzzle-4x4-play-v0 | 65 $_{\pm 4}$ | 0 $_{\pm 0}$ | 0 $_{\pm 0}$ | 0 $_{\pm 0}$ | 0 $_{\pm 0}$ | 0 $_{\pm 0}$ |

representation learning methods and the original representations on most tasks. Notably, our method performs particularly well on `cube` tasks, in some cases improving performance by $3\times$ compared to the "original" baseline (Table 6). Moreover, unlike previous algorithms such as VIB, VIP, and TRA, our method shows the least sensitivity to the choice of downstream offline GCRL algorithms.

### 6.2.2 RESULTS ON PIXEL-BASED TASKS

Next, we present the comparison results on pixel-based tasks in Table 2. We use GCIVL as the base offline GCRL algorithm, given its strongest average performance in the state-based case. As in Section 6.2.1, the results suggest that our dual goal representation outperforms the other five alternative representations (or at least performs equally well) on five out of seven tasks.

**Negative results.** However, we found that no representation learning methods (including ours), achieve non-zero performance on `visual-puzzle` tasks (Table 2). Given their reasonable performance on the equivalent state-based `puzzle` tasks (Table 1), this failure is likely related to handling visual observations.

We hypothesize that the failure stems from differences between early and late fusion of visual state and goal observations. Typically, a vanilla goal-conditioned policy $\pi(a \mid s, g)$ first concatenates $s$ and $g$ along the depth dimension (forming a $64 \times 64 \times (3+3)$-sized image) and passes it to a convolutional neural network (this is of-

Table 3: **Early vs. late fusion.**

| Environment | Early | Late |
|---|---|---|
| visual-puzzle-3x3 | 22 $_{\pm 2}$ | 0 $_{\pm 0}$ |
| visual-puzzle-4x4 | 65 $_{\pm 4}$ | 0 $_{\pm 0}$ |

ten called "early fusion"). However, when using a representation-conditioned policy $\pi(a \mid s, \varphi(g))$, we cannot directly use early fusion as $g$ needs to be processed separately. The problem is that, early fusion generally leads to better performance in visual robotics tasks (Walsman et al., 2019; Huang et al., 2024). To verify this, we compare the performance of *vanilla* GCIVL with early and late fusion. Table 3 shows the results on `puzzle-{3x3, 4x4}`, suggesting that the use of late fusion is in-

deed likely the cause of the poor performance of representation learning-based approaches on these tasks. Given this evidence, we believe this failure may be addressed by modifying the representation function to be aware of the current state as well, which we leave for future work.

### 6.3 Q&As

In this section, we present additional discussions and ablations through the following Q&As.

**Q: Are dual goal representations indeed robust to noise?**

**A:** Our theory (Theorem 3.2) suggests that dual goal representations can, in principle, be invariant to exogenous noise. To empirically verify this property in practice, we evaluate the original and dual goal representations in three "noisy" environments (with GCIVL as the downstream GCRL algorithm), where we add Gaussian noise to the evaluation goals. This assesses the robustness of the two representations to out-of-distribution goals at test time. Figure 4 presents the comparison results. The results tell us that our dual representation indeed exhibits better robustness than the original one, as suggested by the theory.

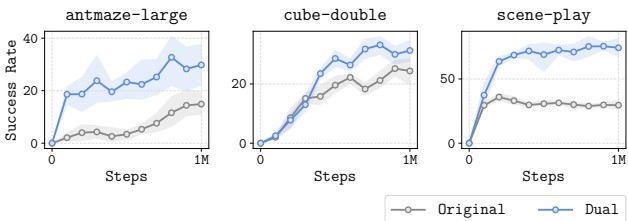

Figure 4: **Dual representations are robust to noise.**

**Q: Can we use other aggregation (parameterization) functions for dual representations?**

Table 4: **Dual representations with different parameterizations.**

| Environment | Dual (Symmetric) | Dual (Inner Product) |
|---|---|---|
| pointmaze-medium-navigate-v0 | $73_{\pm 13}$ | $76_{\pm 3}$ |
| pointmaze-large-navigate-v0 | $43_{\pm 11}$ | $46_{\pm 8}$ |
| antmaze-medium-navigate-v0 | $77_{\pm 7}$ | $75_{\pm 5}$ |
| antmaze-large-navigate-v0 | $23_{\pm 1}$ | $35_{\pm 7}$ |
| antmaze-giant-navigate-v0 | $0_{\pm 0}$ | $1_{\pm 0}$ |
| humanoidmaze-medium-navigate-v0 | $23_{\pm 2}$ | $30_{\pm 4}$ |
| humanoidmaze-large-navigate-v0 | $3_{\pm 1}$ | $3_{\pm 2}$ |
| antsoccer-arena-navigate-v0 | $30_{\pm 3}$ | $31_{\pm 3}$ |
| cube-single-play-v0 | $66_{\pm 4}$ | $90_{\pm 4}$ |
| cube-double-play-v0 | $48_{\pm 6}$ | $61_{\pm 4}$ |
| scene-play-v0 | $82_{\pm 8}$ | $71_{\pm 6}$ |
| puzzle-3x3-play-v0 | $5_{\pm 1}$ | $5_{\pm 0}$ |
| puzzle-4x4-play-v0 | $24_{\pm 1}$ | $23_{\pm 3}$ |
| Average | $38_{\pm 1}$ | $42_{\pm 1}$ |

**A:** In Section 4, we approximate a functional using the inner product aggregation function, $f(\psi(s), \varphi(g)) = \psi(s)^\top \varphi(g)$. However, this is not the only possible option. For example, we can use other parameterizations, such as the symmetric norm (metric) parameterization, $f(\psi(s), \varphi(g)) = \|\varphi(s) - \varphi(g)\|_2$ with $\psi = \varphi$. In particular, if we employ this metric parameterization, our dual representations become similar to previous temporal metric embedding methods (Sermanet et al., 2018; Ma et al., 2023; Park et al., 2024a). To understand how different aggregation functions affect performance, we evaluate dual goal representations with the two parameterizations above (with GCIVL as the downstream GCRL algorithm), and present the results in Table 4. The results show that our inner product representation generally leads to better performance than the metric one when used as a goal representation. We believe this is likely due to the universality of the inner product function (Section 4), whereas the metric parameterization is provably not universal and thus potentially less expressive in representing goals (though this parameterization can still be useful for other purposes; see Ma et al. (2023); Park et al. (2024a)).

**Q: Can't we directly extract a policy from the temporal distance function $f(\psi(s), \varphi(g))$?**

**A:** In our offline GCRL recipe (Algorithm 1), we train *two* goal-conditioned value functions: $f(\psi(s), \varphi(g))$ for approximating temporal distances to learn a dual representation, and $V^{\mathrm{GCRL}}(s, \varphi(g))$ for performing downstream GCRL. One might wonder whether we can directly extract a goal-conditioned policy from the former function $f$, without having a separate downstream GCRL loop. This is indeed possible ("Dual (Direct)" in Table 5), but it generally leads to worse performance (except for `antmaze-large`). The reason is that $f$ has a structured inner product

Table 5: **Directly extracting a policy from the temporal distance function often degrades performance.** In this experiment, we use GCIVL for both $f$ and $V^{\mathrm{GCRL}}$ to enable an apples-to-apples comparison.

| Environment | Dual (Direct) | Dual |
|---|---|---|
| `antmaze-large` | $58_{\pm 6}$ | $28_{\pm 2}$ |
| `humanoidmaze-medium` | $17_{\pm 3}$ | $28_{\pm 2}$ |
| `antsoccer-arena` | $16_{\pm 4}$ | $21_{\pm 2}$ |
| `cube-double` | $42_{\pm 4}$ | $49_{\pm 3}$ |
| `scene-play` | $50_{\pm 4}$ | $71_{\pm 6}$ |
| `puzzle-4x4` | $13_{\pm 1}$ | $24_{\pm 3}$ |

parameterization and is relatively less expressive (although it is theoretically universal) to extract a goal-conditioned policy than the *monolithic* $V^{\mathrm{GCRL}}(s, \varphi(g))$ function, which has no structural constraints. In other words, the inner product parameterization is expressive enough to learn a good goal *representation* but may not be enough to extract a *policy* (*e.g.*, in `antmaze`, it is sufficient to focus on the $x$-$y$ position of the agent to learn a good goal representation, but we need to know the full joint angles to train a control policy), and thus we often benefit from a separate downstream GCRL algorithm with a different, more expressive value function.

## 7 CLOSING REMARKS

Originally, the main idea in this work was inspired by a common slogan in mathematics: "an object is uniquely determined by its relations with every other object." Building on top of this high-level idea, we introduced the concept of a dual goal representation, studied its theoretical properties (sufficiency and noise invariance), proposed a practical goal representation learning recipe, and demonstrated its empirical performance. We hope that the ideas introduced in this work inspire future research on state and goal representation learning in reinforcement learning and sequential decision making.

## ACKNOWLEDGMENT

This work was partly supported by the Korea Foundation for Advanced Studies (KFAS), ONR N00014-22-1-2773, and AFOSR FA9550-22-1-0273. This research used the Savio computational cluster resource provided by the Berkeley Research Computing program at UC Berkeley.

## REPRODUCIBILITY STATEMENT

We provide the code and instructions in the supplementary material and describe the full implementation details in Section D.

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

## A PROOFS

**Theorem A.1** (Sufficiency of Dual Goal Representations). *Let $\mathcal{M} = (\mathcal{S}, \mathcal{A}, p)$ be a CMP and $\varphi^\vee$ be its dual goal representation function. Then, there exists a deterministic policy $\pi^\vee : \mathcal{S} \times \mathcal{S}^\vee \to \mathcal{A}$ that takes a dual goal representation as input, such that its induced policy $\tilde{\pi} : \mathcal{S} \times \mathcal{S} \to \mathcal{A}$ defined as $\tilde{\pi}(s, g) := \pi^\vee(s, \varphi^\vee(g))$ satisfies $V^{\tilde{\pi}}(s, g) = V^*(s, g)$ for all $s, g \in \mathcal{S}$.*

*Proof.* We define $\pi^\vee : \mathcal{S} \times \mathcal{S}^\vee \to \mathcal{A}$ as

$$\pi^\vee(s, f) := \arg\max_{a \in \mathcal{A}} \mathbb{E}_{s' \sim p(s'|s,a)}[\gamma^{f(s')}]$$

for $s \in \mathcal{S}$ and $f \in \mathcal{S}^\vee$. Then, for any $s, g \in \mathcal{S}$, we have

$$\begin{aligned}
\tilde{\pi}(s, g) &= \pi^\vee(s, \varphi^\vee(g)) \\
&= \arg\max_{a \in \mathcal{A}} \mathbb{E}_{s' \sim p(s'|s,a)}[\gamma^{\varphi^\vee(g)(s')}] \\
&= \arg\max_{a \in \mathcal{A}} \mathbb{E}_{s' \sim p(s'|s,a)}[V^*(s', g)] \\
&= \arg\max_{a \in \mathcal{A}} Q^*(s, a, g).
\end{aligned}$$

Invoking the standard result in RL theory (Sutton & Barto, 2005), we conclude that $V^{\tilde{\pi}}(s, g) = V^*(s, g)$. $\qquad\square$

**Theorem A.2** (Noise Invariance of Dual Goal Representations). *Let $\mathcal{M} = (\mathcal{S}, \mathcal{Z}, \mathcal{A}, p, p^e, p^\ell)$ be a BCMP and $\varphi^\vee$ be its dual goal representation function. Let $g_1, g_2 \in \mathcal{S}$ be two goal observations from the same latent state; i.e., $p^\ell(g_1) = p^\ell(g_2)$. Then, they have the same dual goal representation; i.e., $\varphi^\vee(g_1) = \varphi^\vee(g_2)$.*

*Proof.* We need to show $\varphi^\vee(g_1)(s) = \varphi^\vee(g_2)(s)$ for all $s \in \mathcal{S}$. This follows from

$$\begin{aligned}
\varphi^\vee(g_1)(s) &= \log_\gamma V^*(s, g_1) \\
&= \log_\gamma (\max_\pi V^\pi(s, g_1)) \\
&= \log_\gamma \left( \max_\pi \mathbb{E}_{\tau \sim p^\pi(\tau|s_0=s)} \left[ \sum_{t=0}^\infty \gamma^t r^\ell(s_t, g_1) \right] \right) \\
&= \log_\gamma \left( \max_\pi \mathbb{E}_{\tau \sim p^\pi(\tau|s_0=s)} \left[ \sum_{t=0}^\infty \gamma^t \mathbb{I}(p^\ell(s_t) = p^\ell(g_1)) \right] \right) \\
&= \log_\gamma \left( \max_\pi \mathbb{E}_{\tau \sim p^\pi(\tau|s_0=s)} \left[ \sum_{t=0}^\infty \gamma^t \mathbb{I}(p^\ell(s_t) = p^\ell(g_2)) \right] \right) \\
&= \log_\gamma \left( \max_\pi \mathbb{E}_{\tau \sim p^\pi(\tau|s_0=s)} \left[ \sum_{t=0}^\infty \gamma^t r^\ell(s_t, g_2) \right] \right) \\
&= \log_\gamma (\max_\pi V^\pi(s, g_2)) \\
&= \log_\gamma V^*(s, g_2) \\
&= \varphi^\vee(g_2)(s),
\end{aligned}$$

where the $\max$ is taken over the set of all non-goal-conditioned policies $\pi : \mathcal{S} \to \Delta(\mathcal{A})$. $\qquad\square$

## B ALGORITHMS

In this section, we describe the representation learning methods and downstream offline GCRL algorithms considered in our experiments.

### B.1 PRIOR REPRESENTATION LEARNING ALGORITHMS

**VIB (Shah et al., 2021; Park et al., 2023).** VIB trains a goal representation based on information bottleneck (Tishby et al., 1999; Alemi et al., 2017). In our experiments, a VIB representation $\varphi$ is trained directly via the value loss in the downstream GCRL algorithm (except for GCFBC, where we train it via the actor loss). Specifically, we model a downstream value function as $V^{\mathrm{GCRL}}(s, \varphi(g))$, where $\varphi(g)$ denotes a stochastic latent vector sampled from $\mathcal{N}(\mu(g), \Sigma(g))$ with $\mu$ being a mean network and $\Sigma$ being a diagonal covariance network. We train this stochastic latent-parameterized value network using the downstream GCRL loss with an additional KL loss term, $\beta D_{\mathrm{KL}}(\mathcal{N}(\mu(g), \Sigma(g)) \,\|\, \mathcal{N}(0, I))$, to regularize the posterior distribution with a coefficient $\beta$. In our experiments, we find that low $\beta$ values generally work the best, and select the best $\beta$ from $\{0.001, 0.003\}$ for each environment and downstream GCRL algorithm.

**TRA (Myers et al., 2025).** TRA trains a goal representation via a symmetric InfoNCE variant (Myers et al., 2025) of the CRL loss (Eysenbach et al., 2022). We then use this representation to parameterize a downstream GCRL policy and downstream value functions. In the original paper, the TRA representation also gets gradients from the downstream GCRL loss. We perform a hyperparameter sweep on this choice, and enable gradient flow for GCFBC and disable it for GCIVL and CRL.

**BYOL-$\gamma$ (Lawson et al., 2025).** BYOL-$\gamma$ trains a representation based on a temporal self-predictive representation loss (Schwarzer et al., 2021). We use the bidirectional variant in the paper, and use the representation $\varphi$ to parameterize goals in downstream policy and value networks. We do not use $\varphi$ to represent states to enable apples-to-apples comparisons with other goal representation learning algorithms. As in TRA, we also perform a hyperparameter sweep on gradient flows from the downstream GCRL loss to the representation, and enable it for GCFBC and disable it for GCIVL and CRL.

**VIP (Ma et al., 2023).** VIP trains a representation via a metric embedding loss (Sikchi et al., 2024). Specifically, it parameterizes the value function as $V(s, g) = -\|\varphi(s) - \varphi(g)\|_2$, and trains $\varphi$ with a dual RL loss function (Ma et al., 2022; 2023; Sikchi et al., 2024). As with other methods, we use $\varphi$ to represent (only) goals in downstream policy and value networks.

### B.2 DOWNSTREAM OFFLINE GCRL ALGORITHMS

In our experiments, we consider three downstream offline goal-conditioned algorithms: GCIVL, and CRL, and GCFBC. For GCIVL and CRL, we use the original implementations of OGBench (Park et al., 2025a), and for GCFBC, we use the implementation by Park et al. (2025b). We refer to these works for the full descriptions of these algorithms.

## C  ADDITIONAL RESULTS

We provide results on state-based tasks with different downstream GCRL algorithms (CRL and GCFBC) in Tables 6 and 7.

Table 6: **Results on state-based tasks with CRL as the downstream algorithm.**

| Environment | Orig | VIB | VIP | TRA | BYOL-$\gamma$ | Dual |
|---|---|---|---|---|---|---|
| pointmaze-medium-navigate-v0 | $30_{\pm 7}$ | $0_{\pm 0}$ | $55_{\pm 7}$ | $32_{\pm 2}$ | $49_{\pm 6}$ | $33_{\pm 1}$ |
| pointmaze-large-navigate-v0 | $46_{\pm 12}$ | $0_{\pm 0}$ | $50_{\pm 5}$ | $35_{\pm 10}$ | $38_{\pm 11}$ | $39_{\pm 12}$ |
| antmaze-medium-navigate-v0 | $95_{\pm 2}$ | $10_{\pm 1}$ | $94_{\pm 0}$ | $77_{\pm 5}$ | $87_{\pm 3}$ | $93_{\pm 3}$ |
| antmaze-large-navigate-v0 | $87_{\pm 7}$ | $8_{\pm 1}$ | $64_{\pm 10}$ | $71_{\pm 14}$ | $77_{\pm 4}$ | $87_{\pm 2}$ |
| antmaze-giant-navigate-v0 | $11_{\pm 3}$ | $0_{\pm 0}$ | $4_{\pm 1}$ | $3_{\pm 1}$ | $2_{\pm 1}$ | $21_{\pm 4}$ |
| humanoidmaze-medium-navigate-v0 | $57_{\pm 2}$ | $3_{\pm 1}$ | $43_{\pm 1}$ | $57_{\pm 1}$ | $46_{\pm 1}$ | $57_{\pm 4}$ |
| humanoidmaze-large-navigate-v0 | $20_{\pm 5}$ | $1_{\pm 0}$ | $13_{\pm 3}$ | $31_{\pm 3}$ | $22_{\pm 7}$ | $18_{\pm 4}$ |
| antsoccer-arena-navigate-v0 | $22_{\pm 2}$ | $1_{\pm 0}$ | $3_{\pm 1}$ | $7_{\pm 2}$ | $9_{\pm 2}$ | $19_{\pm 3}$ |
| cube-single-play-v0 | $20_{\pm 3}$ | $7_{\pm 2}$ | $42_{\pm 6}$ | $17_{\pm 3}$ | $29_{\pm 5}$ | $60_{\pm 1}$ |
| cube-double-play-v0 | $11_{\pm 3}$ | $2_{\pm 1}$ | $4_{\pm 1}$ | $5_{\pm 2}$ | $3_{\pm 2}$ | $24_{\pm 5}$ |
| scene-play-v0 | $20_{\pm 2}$ | $4_{\pm 1}$ | $18_{\pm 3}$ | $22_{\pm 5}$ | $33_{\pm 2}$ | $44_{\pm 5}$ |
| puzzle-3x3-play-v0 | $4_{\pm 1}$ | $2_{\pm 1}$ | $3_{\pm 1}$ | $3_{\pm 0}$ | $0_{\pm 0}$ | $6_{\pm 1}$ |
| puzzle-4x4-play-v0 | $0_{\pm 0}$ | $0_{\pm 0}$ | $0_{\pm 0}$ | $0_{\pm 0}$ | $0_{\pm 0}$ | $2_{\pm 0}$ |
| Average | $33_{\pm 2}$ | $3_{\pm 0}$ | $30_{\pm 1}$ | $28_{\pm 1}$ | $30_{\pm 1}$ | $39_{\pm 1}$ |

Table 7: **Results on state-based tasks with GCFBC as the downstream algorithm.**

| Environment | Orig | VIB | VIP | TRA | BYOL-$\gamma$ | Dual |
|---|---|---|---|---|---|---|
| pointmaze-medium-navigate-v0 | $63_{\pm 3}$ | $48_{\pm 4}$ | $53_{\pm 8}$ | $77_{\pm 4}$ | $77_{\pm 2}$ | $73_{\pm 3}$ |
| pointmaze-large-navigate-v0 | $70_{\pm 4}$ | $69_{\pm 2}$ | $65_{\pm 5}$ | $76_{\pm 4}$ | $75_{\pm 4}$ | $66_{\pm 7}$ |
| antmaze-medium-navigate-v0 | $44_{\pm 1}$ | $10_{\pm 2}$ | $18_{\pm 2}$ | $56_{\pm 5}$ | $51_{\pm 4}$ | $53_{\pm 9}$ |
| antmaze-large-navigate-v0 | $20_{\pm 1}$ | $2_{\pm 1}$ | $7_{\pm 3}$ | $29_{\pm 2}$ | $29_{\pm 4}$ | $31_{\pm 5}$ |
| antmaze-giant-navigate-v0 | $0_{\pm 0}$ | $0_{\pm 0}$ | $0_{\pm 0}$ | $0_{\pm 0}$ | $0_{\pm 0}$ | $0_{\pm 0}$ |
| humanoidmaze-medium-navigate-v0 | $8_{\pm 1}$ | $3_{\pm 1}$ | $5_{\pm 1}$ | $9_{\pm 1}$ | $8_{\pm 2}$ | $8_{\pm 2}$ |
| humanoidmaze-large-navigate-v0 | $1_{\pm 1}$ | $0_{\pm 0}$ | $0_{\pm 0}$ | $1_{\pm 0}$ | $1_{\pm 0}$ | $1_{\pm 0}$ |
| antsoccer-arena-navigate-v0 | $16_{\pm 1}$ | $3_{\pm 1}$ | $3_{\pm 0}$ | $11_{\pm 1}$ | $12_{\pm 1}$ | $17_{\pm 2}$ |
| cube-single-play-v0 | $12_{\pm 1}$ | $10_{\pm 2}$ | $14_{\pm 1}$ | $12_{\pm 3}$ | $10_{\pm 1}$ | $20_{\pm 2}$ |
| cube-double-play-v0 | $3_{\pm 0}$ | $3_{\pm 1}$ | $3_{\pm 1}$ | $3_{\pm 1}$ | $2_{\pm 1}$ | $5_{\pm 1}$ |
| scene-play-v0 | $19_{\pm 4}$ | $6_{\pm 1}$ | $18_{\pm 4}$ | $20_{\pm 5}$ | $13_{\pm 2}$ | $27_{\pm 3}$ |
| puzzle-3x3-play-v0 | $3_{\pm 1}$ | $2_{\pm 1}$ | $2_{\pm 0}$ | $2_{\pm 0}$ | $2_{\pm 1}$ | $2_{\pm 0}$ |
| puzzle-4x4-play-v0 | $2_{\pm 1}$ | $0_{\pm 0}$ | $0_{\pm 0}$ | $1_{\pm 0}$ | $0_{\pm 0}$ | $0_{\pm 0}$ |
| Average | $20_{\pm 0}$ | $12_{\pm 0}$ | $14_{\pm 1}$ | $23_{\pm 1}$ | $22_{\pm 1}$ | $23_{\pm 1}$ |

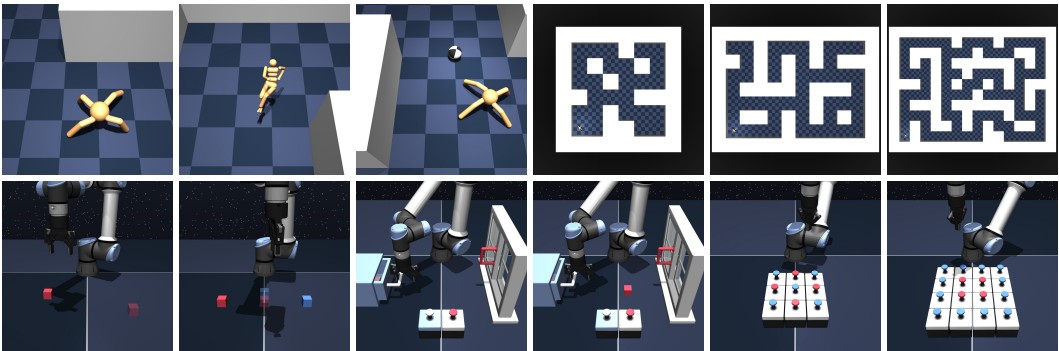

Figure 5: **OGBench environments.** OGBench provides diverse state- and pixel-based goal-conditioned tasks across robotic navigation and manipulation. This figure is taken from Park et al. (2025c).

# D    IMPLEMENTATION DETAILS

Our method and baselines are implemented on top of the default implementations given in the OG-Bench codebase (Park et al., 2025a). We provide the code and instructions at `https://github.com/deepindermann/dual-goal-representations`.

## D.1    DISCRETE PUZZLE EXPERIMENTS

**Tasks and datasets.** In Section 6.1, we employ a discrete "Lights Out" puzzle environment. In this environment, a state consists of $n_x \times n_y$ binary button states (*i.e.*, $\mathcal{S} = \{0,1\}^{n_x n_y}$), and an action corresponds to pressing a button (*i.e.*, $\mathcal{A} = \{0, 1, \ldots, n_x n_y - 1\}$). Whenever the agent presses a button, the colors of that button and the adjacent ones get toggled. We employ two different puzzle sizes, $(n_x, n_y) = (4, 5)$ and $(4, 6)$, which we call `puzzle-4x5` and `puzzle-4x6`, respectively. At test time, we evaluate the agent with the five pre-defined state-goal configurations used in OG-Bench (Park et al., 2025a), where the maximum evaluation episode length is set to $n_x n_y$. For datasets, we employ 40000 length-25 trajectories generated by a uniform random policy.

**Training.** To compute dual representations, we first randomly sample 64 states, run breadth-first search to compute temporal distances, and use the list of temporal distances from the 64 states as a dual representation. For downstream goal-conditioned DQN, we use the $-1$-0 sparse reward function (see Section D.2). We train agents for 1M steps, and evaluate the $\arg\max$ DQN policies using 15 episodes for each of the five evaluation tasks.

**Hyperparameters.** We provide the list of hyperparameters in Table 8. In the table, $(p_{\text{cur}}^{\mathcal{D}}, p_{\text{geom}}^{\mathcal{D}}, p_{\text{traj}}^{\mathcal{D}}, p_{\text{rand}}^{\mathcal{D}})$ denotes the hindsight relabeling ratio tuple defined in Park et al. (2025a).

## D.2    OGBENCH EXPERIMENTS

**Visual encoders.** In pixel-based tasks, there are several design choices regarding visual encoders (*e.g.*, whether to share state or goal encoders across different networks, how to flow gradients, etc.). We test various variants regarding these choices and find the following configuration to work best. For the original representation baseline, we use separate "early fusion" (see Section 6.2.2) encoders for each of the policy and value network. For other goal representation methods, we share a single state visual encoder and a single goal visual encoder for all networks (with "late fusion"), and they get gradients from all downstream GCRL losses as well as the representation learning loss. However, we do not propagate downstream GCRL gradients through the representation function $\varphi$ (except for VIB, which is designed to be trained with the downstream value loss). We refer to our code for the full details about gradient flows.

**Metrics.** Following OGBench (Park et al., 2025a), we report the average performance over the last three evaluation epochs in tables (*i.e.*, over 800K, 900K, and 1M gradient steps for state-based tasks and over 300K, 400K, and 500K steps for pixel-based tasks). We use 50 (state-based) or 15 (pixel-based) episodes for each of the five evaluation goals.

**Hyperparameters.** We provide the list of hyperparameters in Table 9. Following OGBench (Park et al., 2025a), we apply layer normalization (Ba et al., 2016) to all networks except for the policies in GCIVL and CRL.

Table 8: **Hyperparameters for discrete puzzle experiments.**

| Hyperparameter | Value |
|---|---|
| Gradient steps | $10^6$ (state-based) |
| Optimizer | Adam (Kingma & Ba, 2015) |
| Learning rate | 0.0001 |
| Batch size | 1024 |
| MLP size | $[1024, 1024, 1024, 1024]$ |
| Nonlinearity | GELU (Hendrycks & Gimpel, 2016) |
| Layer normalization | True |
| Target network update rate | 0.005 |
| Discount factor $\gamma$ | 0.95 |
| DQN $(p_{\text{cur}}^{\mathcal{D}}, p_{\text{geom}}^{\mathcal{D}}, p_{\text{traj}}^{\mathcal{D}}, p_{\text{rand}}^{\mathcal{D}})$ ratio | $(0.2, 0, 0.5, 0.3)$ |

Table 9: **Hyperparameters for OGBench experiments.**

| Hyperparameter | Value |
|---|---|
| Gradient steps | $10^6$ (state-based), $5 \times 10^5$ (pixel-based) |
| Optimizer | Adam (Kingma & Ba, 2015) |
| Learning rate | 0.0003 |
| Batch size | 1024 (state-based), 256 (pixel-based) |
| MLP size | $[512, 512, 512]$ |
| Nonlinearity | GELU (Hendrycks & Gimpel, 2016) |
| Target network update rate | 0.005 |
| Discount factor $\gamma$ | 0.99 (default), 0.995 (`humanoidmaze`, `antmaze-giant`) |
| Goal representation dimensionality $N$ | 256 (default), 64 (`pointmaze`) |
| Dual representation's GCIQL expectile $\kappa$ | 0.7 (default), 0.9 (`navigate`) |
| VIB $\beta$ | 0.001 (default), 0.003 (`pointmaze`, `antmaze-large`) |
| Representation $(p_{\text{cur}}^{\mathcal{D}}, p_{\text{geom}}^{\mathcal{D}}, p_{\text{traj}}^{\mathcal{D}}, p_{\text{rand}}^{\mathcal{D}})$ ratio (Dual) | $(0.2, 0.5, 0, 0.3)$ |
| Representation $(p_{\text{cur}}^{\mathcal{D}}, p_{\text{geom}}^{\mathcal{D}}, p_{\text{traj}}^{\mathcal{D}}, p_{\text{rand}}^{\mathcal{D}})$ ratio (TRA, BYOL-$\gamma$) | $(0, 1, 0, 0)$ |
| Downstream GCIVL expectile | 0.9 |
| Downstream policy extraction hyperparameters | Identical to OGBench (Park et al., 2025a) |
| Downstream policy $(p_{\text{cur}}^{\mathcal{D}}, p_{\text{geom}}^{\mathcal{D}}, p_{\text{traj}}^{\mathcal{D}}, p_{\text{rand}}^{\mathcal{D}})$ ratio | $(0, 0, 1, 0)$ |
| Downstream value $(p_{\text{cur}}^{\mathcal{D}}, p_{\text{geom}}^{\mathcal{D}}, p_{\text{traj}}^{\mathcal{D}}, p_{\text{rand}}^{\mathcal{D}})$ ratio (default) | $(0.2, 0.5, 0, 0.3)$ |
| Downstream value $(p_{\text{cur}}^{\mathcal{D}}, p_{\text{geom}}^{\mathcal{D}}, p_{\text{traj}}^{\mathcal{D}}, p_{\text{rand}}^{\mathcal{D}})$ ratio (CRL) | $(0, 1, 0, 0)$ |
| Goal noise for Figure 4 | 0.3 (`antmaze`), 0.2 (`cube`, `scene`) |
| Visual encoder | `impala_small` (Espeholt et al., 2018; Park et al., 2025a) |
| Image augmentation probability | 0.5 |