# OpenReview forum: "Dual Goal Representations"
_ICLR.cc/2026/Conference — ICLR 2026 Poster_

### Official Review · Reviewer_U6gt · 2025-10-27

**Soundness:** 3
**Presentation:** 2
**Contribution:** 3
**Rating:** 4
**Confidence:** 2

**Summary:**

The paper proposes a new approach to encode a goal state in goal-conditioned reinforcement learning (GCRL). Instead of using the goal state s directly, the work proposes to use a dual representation where the goal is represent as its distance to all other states. It is claimed that this representation has favorable properties which are theoretically proved and verified by the experimental results on simulated environments.

**Strengths:**

The fundamental idea of the work is simple and interesting. Moreover, the experimental results seem to verify that the simple idea works (at least) in simulations.

**Weaknesses:**

**Major:**

 - The idea of dual representation in Figure 1 is simple, but also yields to certain questions such as dimension of the dual representation in the case of a large number of states or in the continuous cases where states are infinite - this should be clarified to serve the readers

 - The concept of "temporal distance" should be clarified. It generally suggests "number of steps from s to g", but it is not defined

 - Things get confusing when states and goal are transformed to the embeddings psi(s) and phi(g) that transform the state to N-dimensional vectors. Close to Eqs. (1) and (2) psi and phi are defined to be the output and the goal heads, but heads of what? Then the distance function is approximated by *implicit q learning* without any explanation why this is done and how it is done? Q-learning provides actor and critic networks, but are they used and how?I sought information from the appendix D.2 without finding any more explanation?

 - The theory is nicely explained, but the gap from the theory to actual implementation seems large?

**Moderate:**

 - Even the simple example in Figs. 2 and 3 remains unclear as the meaning of state is not explained (i.e. what means the 64 random states - any random configuration?)

 - If the proposed algorithm needs pre-training with IQL, then how many interaction steps are needed for IQL? If that is substantial, then is this method really improving over other methods with the same number of steps and average over N runs with random initializations (for example, in Figure 4 are those IQL steps added to Dual results)?

**Minor**

**Questions:**

As the authors see from my comments, I struggled to understand the practical implementation of the proposed method. I hope that the authors clarify me the concepts or explain why my questions/concerns are incorrect? I am RL practitioner and somewhat experienced coder, but I could not implement your method solely based on the manuscript.

---

> ### Author Response · Authors · 2025-11-21
>
> Thank you for the detailed review and constructive feedback on this work. We especially appreciate your detailed clarification questions about the writing. We have rewritten and revised several sections to improve clarity, and have performed an additional experiment to address the question about training steps. We believe that these updates address the major concerns raised in your review, but if there are any remaining questions, feel free to let us know.
>
> * **"Things get confusing when states and goal are transformed to the embeddings psi(s) and phi(g) that transform the state to N-dimensional vectors." / Clarification about the methodology (heads, implicit Q-learning, etc.)**
>
> Thanks a lot for these clarification questions! We have extensively revised the methodology part (Section 4, changes in red) to clarify these points. We answer your individual questions below:
>
> **"Close to Eqs. (1) and (2) psi and phi are defined to be the output and the goal heads, but heads of what?"**: To recall the context, our goal here is to practically approximate the dual representation, which is defined as the set of temporal distances ($\varphi^\vee(g) = [d^\*(s_1, g), d^\*(s_2, g), \ldots, d^\*(s_K, g)]^\top$), in continuous-state environments. To do this, we parameterize the temporal distance function as $d^\*(s, g) = \psi(s)^\top \varphi(g)$ and treat the goal embedding function $\varphi(g)$ as a practical approximation of the set of temporal distances (the "goal head" simply means this function $\varphi(g)$). The intuition is as follows: even though $\varphi(g)$ obtained above is not literally the set of temporal distances, this embedding $\varphi(g)$ contains the full information about the set of temporal distances (in the ideal case where the approximation is tight), since we can recover any $d^\*(s, g)$ by simply computing the inner product between $\varphi(g)$ and $\psi(s)$ for any $s \in \mathcal{S}$. To clarify this, we replaced the term "heads" with "embeddings" and further clarified this intuition in the revised draft (changes in red).
>
> **"Then the distance function is approximated by implicit q learning without any explanation why this is done and how it is done?"**: Thanks for asking this. Following the suggestion, **we have extensively rewritten this section in the updated draft and provided the full details of the implicit Q-learning (IQL) training procedure in the main text.** At a very high level, one may understand IQL as a black-box algorithm to approximate the temporal distance function $d^\*$ in practice (please refer to Section 4 of the revised paper for the full details).
>
> We hope that the revised draft has clarified the training details of our method. Please let us know if you have any additional questions or clarification concerns about this.
>
> * **The theory is nicely explained, but the gap from the theory to actual implementation seems large?**
>
> Indeed, there exists a gap between the "ideal" dual goal representation (Section 3) and the "practical" dual goal representation (Section 4) as we employ two approximations in practice: (1) the use of inner product parameterization to approximate functionals (the first part in Section 4) and (2) the use of goal-conditioned IQL to approximate $d^\*$ (the second part in Section 4). However, we note that these two approximations are **tight** in the following sense: Regarding the first approximation, if the dimensionality of $\psi$ and $\varphi$ is sufficiently large, we can approximate the temporal distance to arbitrary accuracy (by the universality result of [1]), so we can maintain the full information about the temporal distances in this limit. Regarding the second approximation, the IQL paper [2] theoretically proves that we can precisely recover $d^\*$ (or equivalently $V^\*$) when the IQL expectile $\tau$ tends to $1$ with full dataset support.
>
> Hence, even though we employ two approximations in practice, we have a theoretical guarantee that these gaps disappear in the limit (with sufficiently large representation dimensionality, sufficiently large $\tau$, and sufficiently large dataset support). We believe these properties make our "practical" algorithm not too detached from the theory.

---

> > ### Author Response · Authors · 2025-11-21
> >
> > * **Figure 1 requires clarification about dimensions/continuous state spaces.**
> >
> > Thanks for the feedback! Following the suggestion, we've updated the caption of Figure 1 in the revised PDF (changes in red).
> >
> > * **The concept of "temporal distance" should be clarified.**
> >
> > Thanks for the feedback. We previously formally defined the temporal distance in Section 2 (highlighted in red). We'd be happy to further revise the paper if this definition is unclear, and please let us know in that case.
> >
> > * **Even the simple example in Figs. 2 and 3 remains unclear as the meaning of state is not explained (i.e. what means the 64 random states - any random configuration?)**
> >
> > Yes, $64$ random states simply mean $64$ random board configurations. We have clarified this in the revised PDF (changes in red). The full description of a state can be found in Appendix D.1.
> >
> > * **Regarding training steps**
> >
> > Thanks for this clarification question. First, we note that we *jointly* train the representation and downstream GCRL algorithm in parallel, instead of pre-training the representation, which means that we use the *same* number of gradient steps (1M for state-based tasks) for all methods. This ensures a *fair comparison* between different representation learning algorithms in terms of the number of gradient steps.
> >
> > However, to provide an even stronger answer to this question, we performed an additional experiment in the setting of Figure 4, where we trained standard GCIVL for **2M** steps (twice the number of steps), and compared it with GCIVL + dual representation trained for 1M steps.
> >
> > | Task | GCIVL (at 2M) | GCIVL + Dual (ours, at 1M) |
> > |---|---|---|
> > | $\texttt{antmaze-large-navigate-v0}$ | $\mathbf{29.0} \pm 3.2$ | $\mathbf{30.3} \pm 9.0$ |
> > | $\texttt{scene-play-v0}$ | $23.7 \pm 3.0$ | $\mathbf{75.0} \pm 7.0$ |
> > | $\texttt{cube-double-play-v0}$ | $25.3 \pm 3.7$ | $\mathbf{31.5} \pm 2.3$ |
> >
> > The table above compares the performance averaged over the last three epochs with four seeds. The results suggest that even with twice the number of steps, the performance of the original representation still mostly falls behind that of the dual representation. This further highlights the robustness of dual goal representations to noise.
> >
> > ---
> >
> > We would like to thank you again for raising important clarification questions about our work. We believe the clarifications have strengthened the paper. Please let us know if you have any additional concerns or questions. If we have addressed your concerns, we would appreciate it if you could update the score accordingly.
> >
> > ---
> >
> > [1] Park et al., METRA: Scalable Unsupervised RL with Metric-Aware Abstraction (2024) \
> > [2] Kostrikov et al., Offline Reinforcement Learning with Implicit Q-Learning (2022)

---

### Official Review · Reviewer_oPkP · 2025-10-30

**Soundness:** 3
**Presentation:** 3
**Contribution:** 2
**Rating:** 4
**Confidence:** 4

**Summary:**

The paper proposes to use estimated distances between states as the underlying latent representation in goal-conditioned reinforcement learning (GCRL). This way a goal can be implicitly represented by the distances to all other states, akin to a heuristic estimate. The approach is empirically tested in a set of benchmark domains using deep reinforcement learning on top of the learned goal representation.

**Strengths:**

The main strength of the work is that the proposed algorithm performs well in practice compared to other algorithms for GCRL. Concepts are clearly presented and experiments seem rather extensive.

**Weaknesses:**

As mentioned by the authors, several previous approaches estimate distances or metrics for GCRL by embedding states in a latent state space and applying a metric such as the L2-norm in the latent space. The authors propose using the scalar product instead, which seems highly related to the L2-norm since the squared L2-norm equals
\[
\psi(s)^\top\psi(s) + \varphi(g)^\top\varphi(g) - 2 \psi(s)^\top\varphi(g).
\]
Hence the main term for determining the distance (though negated) is precisely the scalar product in the third term. Another difference is that the goal embedding \varphi is potentially different from the embedding \psi used for other states. Though this has potential positive effects such as representing asymmetric distances, applying a metric to vectors from different embedded spaces does not have the same intuitive interpretation. I believe the motivation for these design choices could be explained better in the text.

In general the claim to novelty is a bit weak. Using the embedding for goal representations seems very similar to using the embedding to estimate distance metrics or perform planning. The inner product seems to slightly outperform the L2-norm in Table 4, but this difference is not large enough to account for the improvement in performance compared to other algorithms.

There exists another work in the literature that also learns explicit distance estimates between pairs of states from offline data:

Steccanella and Jonsson, ECML 2022
State Representation Learning for Goal-Conditioned Reinforcement Learning

**Questions:**

Do you actually use a different embedding for goal states as opposed to other states? What is the motivation for using two different embedding functions?

If planning using the learned distance estimates performs poorly, does this not indicate that the learned distance estimates are not very accurate?

---

> ### Author Response · Authors · 2025-11-21
>
> Thank you for the detailed review and constructive feedback on this work. We especially appreciate your questions about the benefits of our asymmetric inner product parameterization compared to other alternatives, such as metric or symmetric inner product ones. We have performed additional experiments to answer this question and provided clarifications for the other points you raised. We believe that these updates address the major concerns raised in your review, but if there are any remaining questions, feel free to let us know.
>
> * **L2 norm vs. inner product parameterization**
>
> Thank you for raising this question. Indeed, there are several ways to parameterize a value function, including (1) symmetric L2 norm ($\\\|\varphi(s) - \varphi(g)\\\|$), (2) asymmetric L2 norm ($\\\|\psi(s) - \varphi(g)\\\|$), (3) symmetric inner product ($\varphi(s)^\top \varphi(g)$), and (4) asymmetric inner product ($\psi(s)^\top \varphi(g)$, **ours**).
>
> The main advantage of (4) (ours) is as follows: **among them, only the asymmetric inner product parameterization is universal** [1], whereas the other three are provably *not* universal (although there exist some relations between these parameterizations, as noted in the review). Here, "universal" means that it can approximate any arbitrary function $f(s, g)$ to an arbitrary accuracy under some mild assumptions. We provide a proof showing that the other parameterizations are *not* universal at the bottom of the response (please feel free to skip it). Intuitively, a norm function is too "restrictive" due to the triangle inequality and its symmetric nature (even when using different embeddings for $s$ and $g$), whereas an (asymmetric) inner product doesn't have such structural constraints. As a result, the other parameterizations might not accurately capture temporal distances due to their limited expressivity, potentially leading to suboptimal representations and performance.
>
> To empirically verify this claim, we experimentally compared the performance of these four parameterizations.
>
> | Parameterization | $\\\|\varphi(s) - \varphi(g)\\\|$ | $\\\|\psi(s) - \varphi(g)\\\|$ | $\varphi(s)^\top \varphi(g)$ | $\psi(s)^\top \varphi(g)$ (ours) |
> |---|---|---|---|---|
> | Average performance (13 tasks) | $38.4 \pm 1.1$ | $39.4 \pm 1.7$ | $2.3 \pm 0.3$ | $\mathbf{42.0} \pm 1.0$ |
>
> The table above summarizes the results, averaging over the full 13 environments with 4 seeds each (**52 seeds in total** for each parameterization; the same setting as Table 4). The results show that the asymmetric inner product parameterization outperforms the other three by a statistically significant margin, which we believe is partly due to its universality.
>
> In addition, to further support our claim about expressivity, we measured the relationship between the performance and accuracy of each of the four representations, where the accuracy is measured by the squared value difference against a learned monolithic value function $V(s, g)$ (which we treat as an "oracle" value function).
>
> | Environment                                | $\\\|\varphi(s) - \varphi(g)\\\|$   | $\\\|\psi(s) - \varphi(g)\\\|$   | $\varphi(s)^\top \varphi(g)$   | $\psi(s)^\top \varphi(g)$ (ours)   |
> |:-------------------------------------------|:------------------------------------|:---------------------------------|:-------------------------------|:----------------------------|
> | $\texttt{antmaze-large-navigate-v0}$       | $1503 \pm 133$                      | $1587 \pm 98$                    | $2739 \pm 242$                 | $\mathbf{867} \pm 141$      |
> | $\texttt{humanoidmaze-medium-navigate-v0}$ | $7466 \pm 132$                      | $7511 \pm 114$                   | $9665 \pm 183$                 | $\mathbf{3193} \pm 89$      |
> | $\texttt{antsoccer-arena-navigate-v0}$     | $862 \pm 23$                        | $852 \pm 24$                     | $3721 \pm 88$                  | $\mathbf{428} \pm 14$       |
> | $\texttt{cube-double-play-v0}$             | $309 \pm 7$                         | $344 \pm 5$                      | $5699 \pm 44$                  | $\mathbf{197} \pm 11$       |
>
> The table above compares the average squared value errors of different parameterizations (**lower is better**) on four environments. The results suggest that our bilinear parameterization yields the lowest value errors on all four tasks. Moreover, the ranking of value errors ((3) < (1) ~= (2) < (4)) coincides with the ranking of performance from the previous table ((3) < (1) ~= (2) < (4); we refer to our response to Reviewer 4LLs for the individual performances on these four tasks). This suggests that the expressivity of a parameterization is indeed correlated with its actual performance, which highlights the importance of using a universal, expressive parameterization such as inner products.

---

> > ### Author Response · Authors · 2025-11-21
> >
> > * **Do you actually use a different embedding for goal states as opposed to other states? What is the motivation for using two different embedding functions?**
> >
> > Thanks for the question. Yes we do, and again, the main reason behind using different embedding functions ($\psi(s)$ and $\varphi(g)$) is to ensure universality. Otherwise, the parameterization could only encode symmetric temporal distances (i.e., $d^\*(s, g) = d^\*(g, s)$), whereas the true temporal distance function is often asymmetric in practice (e.g., dropping a cube is much easier than picking it up). As the table in the previous question shows, the symmetric inner product parameterization ($\varphi(s)^\top \varphi(g)$, the third one) leads to the weakest performance and the highest approximation errors.
> >
> > * **"In general the claim to novelty is a bit weak."**
> >
> > Indeed, there exist several prior works that train state representations using metric embeddings (e.g., TCN, VIP, HILP, etc.). While our algorithm is different from them in that we use the universal inner product parameterization (as opposed to the L2 norm), another important difference is that we propose to use the learned representation as a **goal embedding** to parameterize the policy (i.e., $\pi(a \mid s, \varphi(g))$), whereas these prior works focus on other usages, such as state representation, planning, reward shaping, etc. (L281-285) (please see the next question for more detailed discussions about planning). Our usage of $\varphi$ as a goal representation is justified based on our (to our knowledge) novel theoretical results in Section 3, which constitute one of the main contributions of this work.
> >
> > To summarize, even though our practical representation learning method itself bears similarity to several prior works, we are (to our knowledge) the first to show that dual representations have desirable properties (i.e., sufficiency and noise invariance) **as a goal representation**, and empirically demonstrate its promising performance as a goal representation learning algorithm.
> >
> > * **Using the embedding for goal representations seems very similar to using the embedding to estimate distance metrics or perform planning. / If planning using the learned distance estimates performs poorly, does this not indicate that the learned distance estimates are not very accurate?**
> >
> > We are a bit unsure whether we have correctly interpreted this question, but we think you might be asking how using the embedding as a goal representation is better than/different from using it for planning (please let us know if this is not what you meant). One main difference is that using it as a goal representation allows us to combine our representation independently with **any** off-the-shelf GCRL algorithm, as shown in the paper (we combined dual representations with three different methods: GCIVL, CRL, and GCFBC), whereas planning requires a specific planning algorithm. Also, unlike methods that embed both states and goals into a metric space (with the $\\\|\varphi(s) - \varphi(g)\\\|$ parameterization) [2, 3, 4], our method only requires the goal embedding ($\varphi(g)$). This enables using a more expressive parameterization like inner product, which improves performance and has a better theoretical guarantee (i.e., universality), as discussed above.

---

> > > ### Author Response · Authors · 2025-11-21
> > >
> > > * **Additional related work ("State Representation Learning for Goal-Conditioned Reinforcement Learning")**
> > >
> > > Thanks for pointing out this work! This work learns a state representation using a symmetric L2 norm parameterization, and we believe it falls into the same high-level category as TCN, VIP, and HILP. We have cited and discussed this work in the revised draft.
> > >
> > > * **"Appendix": Proof for the non-universality of the asymmetric norm parameterization**
> > >
> > > Here, we show that the asymmetric norm parameterization $\\\|\varphi(s) - \psi(g)\\\|$ is not universal (note that the symmetric ones are clearly not universal because they cannot model asymmetric functions). First, we recall that universality means that for any $\epsilon > 0$, there exist $n \in \mathbb{N}$, $\varphi: \mathcal{S} \to \mathbb{R}^n$, and $\psi: \mathcal{S} \to \mathbb{R}^n$ such that $|f(s, g) - \varphi(s)^\top \psi(g)| < \epsilon$ for all $s, g \in \mathcal{S}$.
> > >
> > > To show that the asymmetric norm parameterization is not universal, we construct a counterexample as follows. Let $f$ be a non-negative continuous function satisfying (1) $f(s, s) = 0$ for all $s \in \mathcal{S}$,
> > > and (2) $f(x, y) = 1$ and $f(y, x) = 2$ for some $x, y \in \mathcal{S}$. Assume that a universal approximator exists, and let $\varphi$ and $\psi$ be the functions satisfying $|f(s, g) - \\\|\varphi(s) - \psi(g)\\\|| < \epsilon$ for all $s, g \in \mathcal{S}$, for a given $\epsilon > 0$. Then, we have $\\\|\varphi(s) - \psi(s)\\\| < \epsilon$ for all $s \in \mathcal{S}$.
> > > However, we have $2 = f(y, x) < \\\|\varphi(y) - \psi(x)\\\| + \epsilon \leq \\\|\varphi(y) - \psi(y)\\\| + \\\|\psi(y) - \varphi(x)\\\| + \\\|\varphi(x) - \psi(x)\\\| + \epsilon < \\\|\psi(y) - \varphi(x)\\\| + 3\epsilon < f(x, y) + 4\epsilon = 1 + 4\epsilon$,
> > > and setting $\epsilon = 1/4$ leads to a contradiction.
> > >
> > > ---
> > >
> > > We would like to thank you again for raising important questions about our work. We believe the new ablation experiments about the accuracy of different parameterizations have strengthened the paper. Please let us know if you have any additional concerns or questions. If we have addressed your concerns, we would appreciate it if you could update the score accordingly.
> > >
> > > ---
> > >
> > > [1] Park et al., METRA: Scalable Unsupervised RL with Metric-Aware Abstraction (2024) \
> > > [2] Sermanet et al., Time-Contrastive Networks: Self-Supervised Learning from Video (2018) \
> > > [3] Ma et al., VIP: Towards Universal Visual Reward and Representation via Value-Implicit Pre-Training (2022) \
> > > [4] Park et al., Foundation policies with Hilbert representations (2024)

---

> > > ### Comment · Reviewer_oPkP · 2025-11-22
> > > **Clarification about planning**
> > >
> > > A clarification about my question regarding planning: Since you estimate a value function/distance metric $\psi(s)^\top\varphi(g)$, you could directly use the value estimate to perform planning in the task. Namely, if I am in a state, I can apply all actions and see which resulting state takes me closer to the current goal. If the value function approximation is accurate, this should already perform well, and I would not need to learn a policy (hence Theorem 3.1 does not seem necessary). A policy would only be needed if my value estimate is *not* accurate. In the last Q-A before Section 7 you seem to imply that the learned value estimate f is good but not great, and you need to learn an additional value function (presumably using deep learning) to get a better value estimate. This puts in question how good the learned distance metric is and what its value is.
> > >
> > > Also I guess I fail to see how representing a goal as a set of distances is fundamentally different from learning a distance function. Representing a policy as $\pi(s|a,\varphi(g))$ or $\pi(s|a,g)$ will both lead to some parameterized representation of g if you use deep learning, so why is the result better for some parameterized input $\varphi(g)$?

---

> > > > ### Author Response · Authors · 2025-11-22
> > > >
> > > > Thanks for the prompt response and clarification!
> > > >
> > > > **(1)** Your first point seems related to whether we can directly use $\psi(s)^\top \varphi(g)$ for control. This is a great point, and it is indeed possible to use $\psi(s)^\top \varphi(g)$ to perform planning or directly extract a policy (i.e., train a policy to directly maximize $-\psi(s)^\top \varphi(g)$ without a separate value function). As you noted, the ablation in Table 5 addresses this precise point. The results in Table 5 show that the parameterized distance function $\psi(s)^\top \varphi(g)$ is expressive enough to provide a good *goal representation*, but is not accurate enough to directly extract a *policy*. (Here, we treat *direct* policy extraction and planning interchangeably, given that direct policy extraction is a "neural" version of planning. However, please let us know if you specifically meant planning.)
> > > >
> > > > This is mainly because control (i.e., direct policy extraction or planning) generally requires more information than goal representations. For example, in `antmaze`, it is sufficient to focus on the $x$-$y$ position of the agent in terms of a good goal representation, but we need to know the full joint angles to train a control policy. We have further clarified this point in the revised draft (L503-L505).
> > > >
> > > > Regarding how accurate the parameterized value function is ($\psi(s)^\top \varphi(g)$) compared to the monolithic value function ($V(s, g)$), we would like to refer to the second table in [our response above](https://openreview.net/forum?id=aMKFTidLSM&noteId=2tri6FmhQY). We measured the error between the two, and showed that at least our inner-product parameterization is the most accurate among other parameterizations, and that this in turn leads to better performance.
> > > >
> > > > **(2)** Then, one may wonder *why we even want to train a separate (parameterized value-based) goal representation at all*, given that we can simply just use a vanilla policy $\pi(a \mid s, g)$ and vanilla monolithic value function $V(s, g)$, being completely free from expressivity concerns. We believe this is related to your second point.
> > > >
> > > > One main benefit of using a goal representation (over not using it) is that a good goal representation helps filter out control-irrelevant information (cf. Theorem 3.2) and thus improves generalization. In practice, it helps the agent focus on more "important" information, such as the location of the goal state (in navigation tasks) and the position of the object (in manipulation tasks), rather than background noises or less important internal states. Although a neural network (in theory) might potentially be able to learn this eventually (with sufficiently large data and compute), having an auxiliary representation learning objective and *imposing a representation "bottleneck"* is often helpful in practice (note that gradients don't flow into $\varphi$ in $\pi(a \mid s, \varphi(g))$). We empirically show the benefits of dual representations throughout the paper by comparing dual and vanilla representations in both standard (Tables 1, 2, 6, and 7) and noisy (Figure 4) settings. (Also, we note that this general principle is not limited to our work; although the problem setting is different, many representation learning works in RL have found that training a state representation with an auxiliary loss is often helpful in practice for similar reasons [1])
> > > >
> > > > To summarize, we believe an ideal goal representation should *only* contain "necessary *and* sufficient" information for control (L125-130). In this work, we theoretically show that a dual goal representation satisfies (a version of) this desideratum (Theorem 3.1 and Theorem 3.2), and empirically demonstrate that our practical representation learning objective indeed generally improves both performance and test-time generalization (Tables 1, 2, 6, and 7 and Figure 4).
> > > >
> > > > ---
> > > >
> > > > We hope that our response above helps address the concerns about the benefits of goal representations and expressivity. We would like to thank you again for asking the additional important questions. Please feel free to let us know if you have any remaining concerns or questions.
> > > >
> > > > ---
> > > >
> > > > [1] Echchahed et al., A Survey of State Representation Learning for Deep Reinforcement Learning (2025)

---

### Official Review · Reviewer_4LLs · 2025-10-31

**Soundness:** 3
**Presentation:** 3
**Contribution:** 3
**Rating:** 6
**Confidence:** 3

**Summary:**

The paper proposes a way to encode goals for goal-conditioned reinforcement learning called dual goal representations. The method characterizes a goal by the collection of shortest-time distances from every state to that goal. The authors show that this representation is sufficient to recover an optimal goal-reaching policy and is invariant to observation noise that does not affect latent system dynamics. Practically, they approximate this functional by learning a goal-conditioned value model, taking its ``goal head'' as the goal embedding, and then training any downstream offline GCRL algorithm that consumes this embedding. Experiments on OGBench report average improvements over several prior representation learners across 20 state and pixel tasks, plus ablations comparing inner-product vs metric parameterizations and a study on noise robustness.

**Strengths:**

* The dual view, defining a goal by its temporal distances to all states, is elegant and connects clearly to the control objective. The sufficiency and noise-invariance theorems make the idea precise and motivate learning a representation that depends only on dynamics rather than raw observations.

* The recipe, ``learn a goal-conditioned value model, use its goal head as the representation, then plug into standard offline GCRL,'' is straightforward and compatible with multiple downstream algorithms. This lowers integration cost for practitioners.

* On OGBench, the proposed representations are competitive or best on a majority of state-based and several pixel tasks, and the paper investigates parameterization choices, the limits of directly extracting a policy from the distance model, and robustness to test-time goal noise. These analyses help interpret when the approach works.

**Weaknesses:**

* The key invariance result is proved under an Ex-BCMP model with disjoint observation supports per latent state, a strong assumption seldom satisfied by realistic image observations. The paper then validates ``robustness to noise'' by adding Gaussian noise to evaluation goals, which does not convincingly reflect exogenous distractors in observations or dynamics. A more faithful test would use structured nuisance variation tied to the emission model or distractor objects.

* The method compares favorably on several pixel tasks, yet performance on visual puzzles collapses for all representation learners. The paper attributes this to late fusion that is required when goals are embedded, while the ``original'' baseline can use early fusion. This creates a comparison gap that complicates claims about representation quality, since the fusion strategy is entangled with the representation. The suggestion that state-aware representations could resolve this is plausible, but the paper stops short of implementing it.

* The empirical approach largely repurposes a goal-conditioned value model and takes its goal head as a representation, combined with inner-product scoring. While the dual perspective is neat, the implementation feels incremental relative to prior metric and contrastive goal embeddings. The inner-product ``universality'' claim is cited, but the paper does not quantify approximation error or show cases where the metric form fundamentally fails beyond average score tables.

**Questions:**

* How sensitive is the approach to the accuracy of the learned distance surrogate? Please report correlations between predicted distances and true shortest-path steps on tasks where ground truth can be computed, and study how downstream performance varies as you degrade the value model. This could validate the core mechanism that “better distance, better policy.”

* Can you evaluate noise invariance with structured distractors rather than additive goal noise, for example by adding irrelevant moving objects or textured backgrounds in pixel tasks while keeping latent dynamics fixed? This would better mirror the Ex-BCMP setup used in the theorem.

* To address the early vs late fusion confound, can you try a state-aware goal embedding or a lightweight cross-attention block that allows early interaction while keeping a representation bottleneck, then re-run the visual puzzles and other pixel tasks for a fair comparison? Even a small prototype would strengthen the claim.

* Please expand the ablation on representation dimensionality and the choice of inner-product vs metric forms. Are there tasks where the metric version wins once you scale capacity or adjust negative sampling for metric learning baselines, and how do results change with larger N?

---

> ### Author Response · Authors · 2025-11-21
>
> Thank you for the detailed review and constructive feedback on this work. We especially appreciate your questions about the relationship between the accuracy of distance models and performance. We have performed several additional experiments and ablations to answer this question, and provided clarifications to the other points. Please find our response below.
>
>
> * **"How sensitive is the approach to the accuracy of the learned distance surrogate?" / Could you validate "better distance, better policy"?**
>
> Thanks for the question! We conducted an additional experiment to show the relationship between the accuracy of the parameterized distance model and the downstream performance. To do this, we considered the following four different parameterizations: (1) symmetric L2 norm ($\\\|\varphi(s) - \varphi(g)\\\|$), (2) asymmetric L2 norm ($\\\|\psi(s) - \varphi(g)\\\|$), (3) symmetric inner product ($\varphi(s)^\top \varphi(g)$), and (4) asymmetric inner product ($\psi(s)^\top \varphi(g)$, **ours**). Then, we compared their performance and accuracy, where accuracy is measured by the squared value difference against a learned monolithic value function $V(s, g)$ (which we treat as an "oracle" value function).
>
>
> **[Error in distance models ($\downarrow$)]**
>
> | Environment                                | $\\\|\varphi(s) - \varphi(g)\\\|$   | $\\\|\psi(s) - \varphi(g)\\\|$   | $\varphi(s)^\top \varphi(g)$   | $\psi(s)^\top \varphi(g)$ (ours)   |
> |:-------------------------------------------|:------------------------------------|:---------------------------------|:-------------------------------|:----------------------------|
> | $\texttt{antmaze-large-navigate-v0}$       | $1503 \pm 133$                      | $1587 \pm 98$                    | $2739 \pm 242$                 | $\mathbf{867} \pm 141$      |
> | $\texttt{humanoidmaze-medium-navigate-v0}$ | $7466 \pm 132$                      | $7511 \pm 114$                   | $9665 \pm 183$                 | $\mathbf{3193} \pm 89$      |
> | $\texttt{antsoccer-arena-navigate-v0}$     | $862 \pm 23$                        | $852 \pm 24$                     | $3721 \pm 88$                  | $\mathbf{428} \pm 14$       |
> | $\texttt{cube-double-play-v0}$             | $309 \pm 7$                         | $344 \pm 5$                      | $5699 \pm 44$                  | $\mathbf{197} \pm 11$       |
>
> **[Performance ($\uparrow$)]**
>
> | Environment                                | $\\\|\varphi(s) - \varphi(g)\\\|$   | $\\\|\psi(s) - \varphi(g)\\\|$   | $\varphi(s)^\top \varphi(g)$   | $\psi(s)^\top \varphi(g)$ (ours)   |
> |:-------------------------------------------|:------------------------------------|:---------------------------------|:-------------------------------|:----------------------------|
> | $\texttt{antmaze-large-navigate-v0}$       | $23 \pm 1$                          | $33 \pm 4$                       | $3 \pm 1$                      | $\mathbf{35} \pm 7$         |
> | $\texttt{humanoidmaze-medium-navigate-v0}$ | $23 \pm 2$                          | $23 \pm 3$                       | $2 \pm 0$                      | $\mathbf{30} \pm 4$         |
> | $\texttt{antsoccer-arena-navigate-v0}$     | $\mathbf{30} \pm 3$                 | $26 \pm 3$                       | $1 \pm 0$                      | $\mathbf{31} \pm 3$         |
> | $\texttt{cube-double-play-v0}$             | $48 \pm 6$                          | $31 \pm 24$                      | $1 \pm 0$                      | $\mathbf{61} \pm 4$         |
>
> The first table compares the accuracy of different parameterizations, and the second table compares their performance. The results suggest that our bilinear parameterization leads to the lowest value errors and best performance on most tasks. Furthermore, the ranking in terms of value errors ((3) < (1) ~= (2) < (4)) coincides with the ranking in terms of performance from the previous table ((3) < (1) ~= (2) < (4)). This suggests that the expressivity of a parameterization is indeed correlated with its actual performance. In particular, it shows why our inner product parameterization can be particularly beneficial compared to previous metric-based ones.

---

> > ### Author Response · Authors · 2025-11-21
> >
> > * **"To address the early vs late fusion confound, can you try a state-aware goal embedding or a lightweight cross-attention block that allows early interaction while keeping a representation bottleneck, then re-run the visual puzzles and other pixel tasks for a fair comparison?"**
> >
> > First, we would like to mention that all representation learning methods (including ours) use late fusion in pixel-based environments (Table 2), which ensures a *fair comparison* between different representation learning methods. While we suspect that we may improve performance on the `puzzle` tasks if we can somehow incorporate early fusion within the goal representation learning framework (as Table 3 suggests), this requires a non-trivial modification because early fusion (by definition) requires concatenating $s$ and $g$ before computing the representation $\varphi$, where most representation learning methods assume that $\varphi$ only depends on $g$. We note that this issue is not limited to our method: in fact, all other goal representation baselines in Table 2 share the same assumption (or limitation). We believe this issue might be addressed by modifying the objective to be state-aware while limiting the information flow from $s$ to $\varphi$ (e.g., with an information bottleneck), and we leave exploring this (non-trivial) change for future work.

---

> > > ### Author Response · Authors · 2025-11-21
> > >
> > > * **"Please expand the ablation on representation dimensionality and the choice of inner-product vs metric forms."**
> > >
> > > Thanks for the question. Following the suggestion, we evaluated the accuracy and performance of metric and bilinear (i.e., inner-product) representations with different dimensionalities ($N$). We used the same accuracy metric as in our experiment in the first question (i.e., the average value error against a learned monolithic value function).
> > >
> > > **[Error in distance models ($\downarrow$)]**
> > >
> > > | Environment                                | Metric $(N=32)$       | Bilinear (ours) $(N=32)$   | Metric $(N=64)$   | Bilinear (ours) $(N=64)$   | Metric $(N=256)$   | Bilinear (ours) $(N=256)$   |
> > > |:-------------------------------------------|:----------------------|:---------------------------|:------------------|:---------------------------|:-------------------|:----------------------------|
> > > | $\texttt{antmaze-large-navigate-v0}$       | $1470 \pm 61$         | $\mathbf{1280} \pm 164$    | $1484 \pm 89$     | $\mathbf{1064} \pm 114$    | $1557 \pm 56$      | $\mathbf{892} \pm 80$       |
> > > | $\texttt{humanoidmaze-medium-navigate-v0}$ | $7522 \pm 159$        | $\mathbf{4908} \pm 226$    | $7296 \pm 84$     | $\mathbf{4288} \pm 143$    | $7543 \pm 136$     | $\mathbf{3247} \pm 75$      |
> > > | $\texttt{antsoccer-arena-navigate-v0}$     | $848 \pm 22$          | $\mathbf{626} \pm 18$      | $853 \pm 17$      | $\mathbf{583} \pm 12$      | $867 \pm 19$       | $\mathbf{428} \pm 21$       |
> > > | $\texttt{cube-double-play-v0}$             | $\mathbf{337} \pm 16$ | $396 \pm 21$               | $316 \pm 8$       | $\mathbf{282} \pm 15$      | $317 \pm 12$       | $\mathbf{204} \pm 9$        |
> > >
> > > **[Performance ($\uparrow$)]**
> > >
> > > | Environment                                | Metric $(N=32)$     | Bilinear (ours) $(N=32)$   | Metric $(N=64)$     | Bilinear (ours) $(N=64)$   | Metric $(N=256)$    | Bilinear (ours) $(N=256)$   |
> > > |:-------------------------------------------|:--------------------|:---------------------------|:--------------------|:---------------------------|:--------------------|:----------------------------|
> > > | $\texttt{antmaze-large-navigate-v0}$       | $26 \pm 2$          | $\mathbf{42} \pm 6$        | $22 \pm 7$          | $\mathbf{30} \pm 4$        | $17 \pm 3$          | $\mathbf{23} \pm 7$         |
> > > | $\texttt{humanoidmaze-medium-navigate-v0}$ | $\mathbf{21} \pm 3$ | $\mathbf{22} \pm 3$        | $20 \pm 5$          | $\mathbf{23} \pm 5$        | $23 \pm 4$          | $\mathbf{28} \pm 4$         |
> > > | $\texttt{antsoccer-arena-navigate-v0}$     | $\mathbf{20} \pm 3$ | $\mathbf{21} \pm 2$        | $21 \pm 4$          | $\mathbf{25} \pm 3$        | $\mathbf{28} \pm 5$ | $\mathbf{29} \pm 6$         |
> > > | $\texttt{cube-double-play-v0}$             | $\mathbf{53} \pm 6$ | $44 \pm 15$                | $\mathbf{56} \pm 7$ | $\mathbf{58} \pm 5$        | $52 \pm 7$          | $\mathbf{60} \pm 6$         |
> > >
> > > The tables above show both the accuracy and performance of metric and bilinear parameterizations with $N \in \\\{32, 64, 256\\\}$ (we highlight the best results within each value of $N$). First, in terms of accuracy, the table shows that bilinear value errors (ours) generally decrease as $N$ grows. In contrast, metric value errors often plateau even with larger $N$ values. We believe this is due to the limited expressivity of the metric value function (e.g., it is always symmetric and must satisfy structural constraints like the triangle inequality), which hinders fully utilizing the expressivity provided by larger dimensionality. In terms of performance, the table shows that our bilinear representation achieves better performance with larger $N$ (except on `antmaze`, likely because specifying a goal only requires two dimensions), mostly outperforming metric representations, especially when $N$ is large. This highlights the importance of expressive parameterization (e.g., inner product parameterization with a sufficiently large $N$).
> > >
> > > ---
> > >
> > > We would like to thank you again for raising important questions about our work. We believe the new ablation experiments have strengthened the paper. Please let us know if you have any additional concerns or questions. If we have fully addressed your concerns, we would appreciate it if you could update the score accordingly.
> > >
> > > ---
> > >
> > > [1] Efroni et al., Provable RL with Exogenous Distractors via Multistep Inverse Dynamics (2022)

---

### Official Review · Reviewer_4M3V · 2025-11-06

**Soundness:** 3
**Presentation:** 3
**Contribution:** 3
**Rating:** 8
**Confidence:** 4

**Summary:**

The paper introduces dual goal representations for goal-conditioned reinforcement learning (GCRL). The core idea is to represent a goal state not by its own features, but by the set of temporal distances from all other states to itself. The paper theoretically proves that this representation is sufficient for learning an optimal policy and, importantly, is invariant to exogenous noise. While the theory establishes these desirable properties to make this concept practical, the proposed algorithm sidesteps the computational challenge by learning a compact, low-dimensional vector that approximates this representation. A comprehensive set of experiments demonstrates the benefit of this approach, showing that it consistently improves the learning performance of several existing GCRL algorithms across a wide range of tasks.

**Strengths:**

- The paper introduces a novel and well-motivated approach to goal representation based on temporal distances. This "dual" perspective is a strong conceptual contribution with clear potential for applications even beyond the GCRL setting.

- The paper is well-written and clearly structured. The authors effectively build intuition by cleanly separating the ideal theoretical concept from its practical implementation, making the core ideas easy to follow and understand.

- The experimental evaluation is both extensive and convincing. By testing on a diverse suite of environments and integrating their representation with three different downstream GCRL algorithms (GCIVL, CRL, GCFBC), the authors provide strong evidence for the method's general applicability and robustness.

**Weaknesses:**

- A significant gap exists between the theoretical framework, which analyzes an ideal functional $\varphi^v$, and the practical implementation, which learns a finite-dimensional approximation. Is not obvious that the formal guarantees of sufficiency and noise invariance do transfer to the compressed low dimensional learned representation. The justification for the method's effectiveness therefore rests heavily on its strong empirical validation rather than a direct theoretical proof for the practical algorithm.

- The proposed method employs a two-stage architecture: first learning a representation $\varphi(g)$ by training an approximate value function ($\psi(s)^T \varphi(g)$), and then using $\varphi(g)$ to train a separate downstream policy and, in many cases, a new value function (V_GCRL). This design raises significant questions about efficiency and potential redundancy.
While the paper's ablation study (Table 5) shows that direct policy extraction from the initial value function performs worse, the underlying reason for this is not fully explored. It seems that a deeper analysis is needed to justify why the inner product parameterization is expressive enough to yield a powerful representation, but too restrictive for direct policy extraction. Clarifying this would explain why the apparent redundancy of learning a second value function and discarding the initial state encoder $\psi(s)$ is a necessary design choice rather than a limitation.

- A minor fix is that the paper is missing the citation to the paper "State Representation learning for Goal Conditioned Reinforcement Learning" Steccanella et al. that seems relevant and learn a temporal distance representation used for reward shaping.

**Questions:**

See Weaknesses

---

> ### Author Response · Authors · 2025-11-21
>
> Thank you for the detailed and positive review and constructive feedback on this work. We especially appreciate your questions about the gap between theory and practice and the necessity of a separate value function. Please find our response below.
>
> * **A gap between the theoretical framework and practical implementation.**
>
> Indeed, there exists a gap between the "theoretical" dual goal representation (Section 3) and the "practical" dual goal representation (Section 4) as we employ two approximations in practice: (1) the use of inner product parameterization to approximate functionals (the first part in Section 4) and (2) the use of goal-conditioned IQL to approximate $d^\*$ (the second part in Section 4). However, we note that these two approximations are "tight" in the following sense: Regarding the first approximation, if the dimensionality of $\psi$ and $\varphi$ is sufficiently large, we can approximate the temporal distance to arbitrary accuracy (by the universality result of [1]), so we can maintain the full information about the temporal distances in this limit. Regarding the second approximation, the IQL paper [2] theoretically proves that we can precisely recover $d^\*$ (or equivalently $V^\*$) when the IQL expectile $\tau$ tends to $1$ with full dataset support.
>
> Hence, even though we employ two approximations in practice, we have a theoretical guarantee that these gaps disappear in the limit (with sufficiently large representation dimensionality, sufficiently large $\tau$, and sufficiently large dataset support). We believe these properties make our "practical" algorithm not too detached from the theory.
>
> That being said, we agree with the reviewer that, under these approximations, we cannot always guarantee the theoretical properties of dual goal representations in practice. However, as the reviewer points out, our empirical results (Table 1 and Figure 4) suggest that dual goal representations are indeed performant and robust to noise compared to other baselines. We believe that these results serve as empirical evidence that dual representations maintain the desirable properties to some degree in practice.
>
> * **Why do we need to train an additional goal-conditioned value function when we already have a distance function $d^\*(s, g) = \psi(s)^\top \varphi(g)$?**
>
> Thanks for asking this question. First, as Table 5 shows, directly extracting a policy from the learned parameterized distance function $\psi(s)^\top \varphi(g)$ empirically leads to worse performance than training an additional goal-conditioned RL agent with the goal representation $\varphi(g)$. This is mainly because the parameterized value function may be too "lossy" to extract a *policy* directly from it (even though it is universal in theory and is expressive enough to learn a *goal representation* in practice). Intuitively, policy extraction generally requires richer information than goal representations. For example, in `antmaze`, it is sufficient to focus on the $x$-$y$ position of the agent in terms of a good goal representation, but we need to know the full joint angles to train a control policy. We have further clarified this point in the revised draft (changes in red).
>
> * **Additional related work ("State Representation Learning for Goal-Conditioned Reinforcement Learning")**
>
> Thanks for pointing out this work! This work learns a state representation using a symmetric L2 norm parameterization, and we believe it falls into the same high-level category as TCN, VIP, and HILP. We have cited and discussed this work in the revised draft.
>
> ---
>
> We would like to thank you again for raising important questions about our work. We hope that our response has addressed your questions. Please feel free to let us know if you have any additional concerns or questions.
>
> ---
>
> [1] Park et al., METRA: Scalable Unsupervised RL with Metric-Aware Abstraction (2024) \
> [2] Kostrikov et al., Offline Reinforcement Learning with Implicit Q-Learning (2022)

---

### Meta-Review · Area_Chair_Vjud · 2026-01-07

**Summary:**

The paper proposes an approach for goal-conditioned RL where a goal representation is trained to ideally contain the information of distance to every other state from the given state. The idea is that such a goal representation will filter out exogenous noise which can be useful for generalization. This is shown theoretically in a representation-sufficiency result in Theorem 3.1 and in practice is approximated using learning Q function and a universal inner product function.

My main concerns are:
- theoretical gap between theory and experiment seems large. E.g., there is no finite-sample result and there is no guarantee that one will learn an accurate distance function with the given representation even if it is universal. Universal does not imply it will work good in practice (e.g., 2 layer MLP with sigmoid activation function are universal). Reviewers noted this as well.
- the generalization argument also falls through because even though the representation filters out exogenous noise, it only filters out noise on which it was trained. If suddenly the noise was changed to something else, it won't work. This argument which is central to the argument is entirely empirically justified in this paper. This is okay but the authors should say so.
- Robustness to noise is shown using simple Gaussian noise.

For these reason, I recommend this paper for weak accept.

**Reviewer Concerns:**

Reviewers raised the following main concerns

- Gap between theory and experiments is large. Reviewer 4M3V and U6gt express this. I believe these concerns remain despite author's rebuttal.

- Reviewers were also confused why should one not use the learned distance function/value function to take action. Authors have explained that this is empirically worse, and also learning a policy on top of goal representations that filter exogenous noise will generalize better. While this is not theoretically shown, it is empirically established so I think this is partially addressed.

- Reviewers also complained about lack of novelty citing many goal-representation literature. I believe the rebuttal address this.

**Reviewer Scores:**

- Reviewer 4M3V gives a score of 8 and unlikely to change based on the rebuttal.

- Reviewer 4LLs gave a score of 6. Their main concern are contrived simple noise in the experiments and failure in visual setting. The former concern remains unaddressed. Overall, I think reviewer would have kept their score at 6.

- Reviewer oPkP gave a score of 4. Their main concern is lack novelty and I believe authors have somewhat addressed that. However, given that there is a lot of related work, I believe they would have increased their score to no more than 5/6.

- Reviewer U6gt gave a score of 4. Their main concern is large gap between theory and practice as well as whether inner product needs very large dimension. The former concern remains and I believe reviewer would have only increased their score a bit to 5/6.

Overall, this is a borderline paper leaning positive.

---

### Decision · Program_Chairs · 2026-01-26

Accept (Poster)